# Muscle-specific CRISPR/Cas9 dystrophin gene editing ameliorates pathophysiology in a mouse model for Duchenne muscular dystrophy

Niclas E. Bengtsson[1,2], John K. Hall[1,2], Guy L. Odom[1,2], Michael P. Phelps[3], Colin R. Andrus[4,5], R. David Hawkins[4,5], Stephen D. Hauschka[2,6], Joel R. Chamberlain[2,4] & Jeffrey S. Chamberlain[1,2,4,6]

Gene replacement therapies utilizing adeno-associated viral (AAV) vectors hold great promise for treating Duchenne muscular dystrophy (DMD). A related approach uses AAV vectors to edit specific regions of the DMD gene using CRISPR/Cas9. Here we develop multiple approaches for editing the mutation in dystrophic $mdx^{4cv}$ mice using single and dual AAV vector delivery of a muscle-specific Cas9 cassette together with single-guide RNA cassettes and, in one approach, a dystrophin homology region to fully correct the mutation. Muscle-restricted Cas9 expression enables direct editing of the mutation, multi-exon deletion or complete gene correction via homologous recombination in myogenic cells. Treated muscles express dystrophin in up to 70% of the myogenic area and increased force generation following intramuscular delivery. Furthermore, systemic administration of the vectors results in widespread expression of dystrophin in both skeletal and cardiac muscles. Our results demonstrate that AAV-mediated muscle-specific gene editing has significant potential for therapy of neuromuscular disorders.

[1] Department of Neurology, University of Washington, Seattle, Washington 98195-7720, USA. [2] Senator Paul D. Wellstone Muscular Dystrophy Cooperative Research Center, University of Washington, Seattle, Washington 98195-7720, USA. [3] Department of Pathology, University of Washington, Seattle, Washington 98195-7720, USA. [4] Department of Medicine, University of Washington, Seattle, Washington 98195-7720, USA. [5] Department of Genome Sciences, University of Washington, Seattle, Washington 98195-7720, USA. [6] Department of Biochemistry, University of Washington, Seattle, Washington 98195-7720, USA. Correspondence and requests for materials should be addressed to J.S.C. (email: jsc5@uw.edu).

Duchenne muscular dystrophy (DMD) is among the most common human genetic disorders, affecting approximately 1:5,000 newborn males[1,2]. Mutations in the dystrophin (DMD) gene result in loss of expression of both dystrophin and the dystrophin-glycoprotein complex, causing muscle membrane fragility, cycles of necrosis and regeneration and progressive muscle wasting[1,3,4]. A variety of approaches for gene therapy of DMD are in development, many of which take advantage of the ability of vectors derived from adeno-associated virus (AAV) to deliver genes systemically via the vasculature[5,6]. While many AAV vectors display a broad tissue tropism, highly restricted muscle expression can be achieved by using muscle-specific gene regulatory cassettes[7]. Two promising methods involving AAV vectors include gene replacement using micro-dystrophins and direct gene editing using CRISPR/Cas9 (refs 5,6). One limitation of these approaches is the ~5 kb AAV vector packaging limit. Micro-dystrophins that lack non-essential domains can be delivered to dystrophic animals using AAV, halting ongoing necrosis and markedly reducing muscle pathophysiology. However, these ~4 kb micro-dystrophins do not fully restore strength[8–11], whereas direct gene editing could lead to production of larger dystrophins, depending on the specific mutation in a patient's genome[12].

The potential for DMD gene modification using the CRISPR/Cas9 system has previously been demonstrated in patient-derived induced pluripotent stem cells (iPSCs) and murine germline manipulation studies[13,14]. Recent studies also utilized the CRISPR/Cas9 system for in vivo excision of exon 23 of the murine Dmd gene[15–17], which carries a nonsense mutation in the $mdx^{ScSn}$ mouse[18]. However, several features of DMD present significant challenges for widespread development of gene editing strategies. DMD is inherited in an X-linked recessive pattern, and one-third of all cases result from spontaneous new mutations in the 2.2 MB DMD gene[1,2]. Thousands of independent mutations have been found in patients (http://www.dmd.nl), which can involve any of the 79 exons that encode the muscle transcript[7,19]. Consequently, gene editing approaches to treat the majority of patients will require great flexibility. To determine the applicability of this system to a wider range of mutational contexts, we explored multiple gene editing strategies in the $mdx^{4cv}$ mouse model that harbours a nonsense mutation within exon 53 (ref. 20). Importantly, this exon is within a mutational hot spot region spanning exons 45–55 that carries the genetic lesion in ~60% of DMD patients with deletion mutations[21]. Importantly, the $mdx^{4cv}$ model exhibits fewer dystrophin-positive revertant myofibers than the original $mdx^{ScSn}$ strain and has a more progressive phenotype. In contrast to exon 23, excision of exon 53 will not restore an open-reading frame (ORF) to the mRNA; therefore a much larger genomic region containing both exons 52 and 53 must be removed or the mutation itself must be directly targeted. Exon 53 editing is thus an instructive additional Duchenne muscular dystrophy (DMD) target since editing different regions of the enormous DMD locus could generate different results due to effects on pre-messenger RNA (mRNA) splicing and the stability and/or functional properties of modified dystrophins that are not predictable[8]. Here we develop and assess multiple muscle-specific, AAV-CRISPR/Cas9-driven gene editing strategies towards the correction of the Dmd gene in dystrophic $mdx^{4cv}$ mice. Treated muscles display robust and widespread dystrophin expression following both local and systemic delivery, resulting in significant morphometric and pathophysiological amelioration of the dystrophic phenotype. Further, we demonstrate successful and novel in vivo induction of homology-directed repair (HDR)-mediated Dmd gene correction. Our results indicate that AAV-CRISPR/Cas9-mediated gene editing has significant potential for the development of future therapies for DMD.

## Results

**Strategies for Dmd gene correction in $mdx^{4cv}$ mice.** Induction of dystrophin expression was tested following AAV6-mediated delivery of CRISPR/Cas9 components derived from either Streptococcus pyogenes (SpCas9)[22] or Staphylococcus aureus (SaCas9)[23] using dual- or single-vector approaches, respectively (Fig. 1a–e). Cas9 expression was restricted to skeletal and cardiac muscle by use of the muscle-specific CK8 regulatory cassette (RC)[24] to reduce the risk of off-target events in non-muscle cells and to minimize elicitation of an immune response[25,26]. We tested several approaches to either excise exons 52 and 53 (Δ5253; strategy 1) or to directly target the mutation in exon 53 (53*; strategy 2). Due to the ~5 kb packaging limit of AAV we designed dual AAV vectors to work in tandem: a nuclease vector expressing SpCas9 under control of the CK8 RC and a set of targeting vectors containing two single-guide RNA (sgRNA) expression cassettes unique to strategies 1 or 2 (Fig. 1a–e). A variant of strategy 1 relying on CK8-regulated expression of the smaller SaCas9 enabled use of a single vector (Fig. 1a).

The overall approaches used in strategy 1 (Δ5253) are potentially applicable to a majority of DMD patients with mutations affecting one or more exons whose removal via editing would allow production of a mRNA with an ORF. For this, we designed sgRNAs to direct Cas9-mediated DNA cleavage within the introns flanking exons 52–53 (Fig. 1a). Following DNA repair via non-homologous end joining (NHEJ) these would result in deletion of ~45 kb of genomic DNA and 330 bp in the encoded mRNA. Successful deletion with strategy 1 will remove the nonsense mutation and lead to the expression of a dystrophin lacking 110 amino acids in a non-essential portion of the protein (Fig. 1b). Strategy 2 (53*) was developed to target small mutations directly, in this case in exon 53, using two distinct methods. These approaches could be applicable to patients with mutations in exons encoding essential domains of dystrophin, such as the dystroglycan-binding domain[27]. The first approach within strategy 2 relies on the introduction of a 'mutation-corrected' DNA template to allow for potential HDR following Cas9-mediated DNA cleavage, resulting in full-length endogenous dystrophin expression (Fig. 1c,d). In the absence of successful HDR, this approach could still enable dystrophin expression where NHEJ repair of the cleaved exon 53 leads to excision of the nonsense mutation while maintaining an ORF in the resultant mRNA (Fig. 1c,e).

**In vivo editing and gene correction in $mdx^{4cv}$ mice.** Dystrophin gene targeting was initially evaluated in vitro using the T7 endonuclease 1 assay in $mdx^{4cv}$-derived primary dermal fibroblasts. The respective targeting efficiencies for sgRNA-i51 and sgRNA-i53 were 9 and 16%, while a combined targeting efficiency of 8% was observed for the 5′ and 3′ sgRNAs within exon 53 (which due to their close proximity were analysed together; Supplementary Fig. 1). For initial in vivo testing 10–12 week old male $mdx^{4cv}$ mice were injected in the tibialis anterior (TA) muscles with $5 \times 10^{10}$ vector genomes (v.g.) of the AAV6 CK8-nuclease plus targeting vectors and sacrificed at 4 weeks post-injection. In vivo targeting efficiency was estimated via deep sequencing across target regions within the dystrophin gene. For strategy 1 PCR amplification of the genomic DNA region spanning the intron 51–53 target sites revealed low levels of a unique Δ5253 deletion product whose sequence was verified following isolation and cloning (Supplementary Fig. 2). Due to the large size of the genomic deletion,

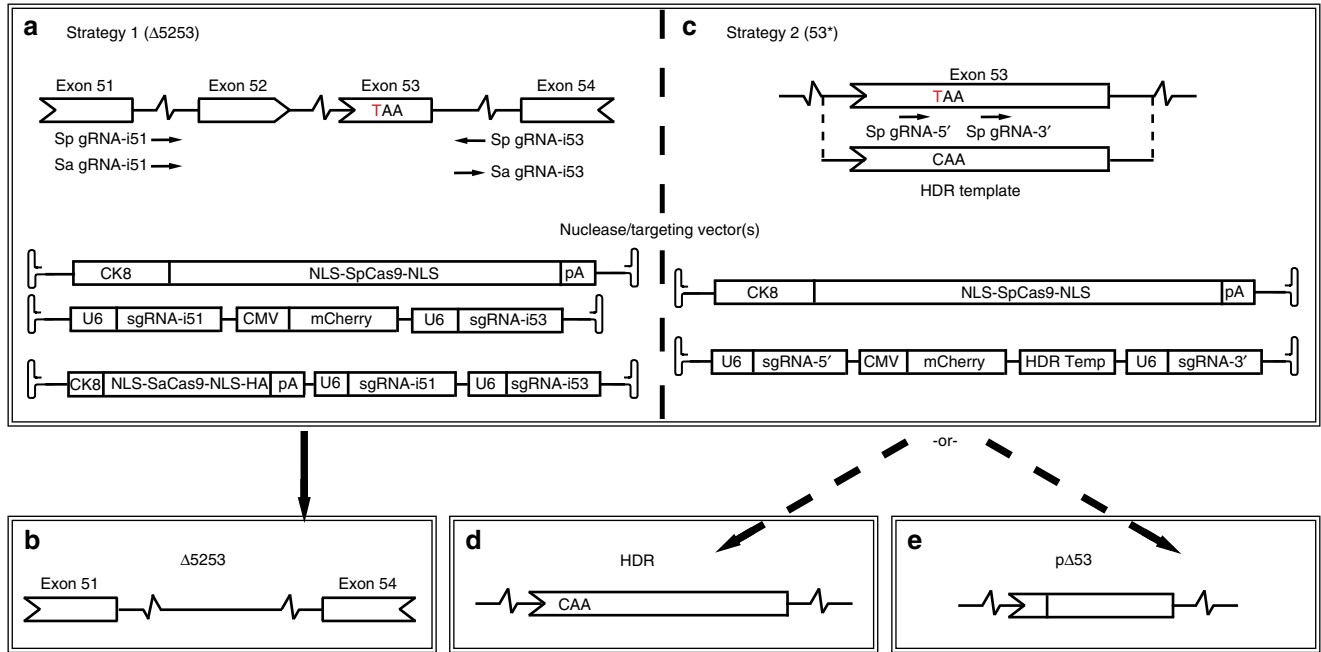

**Figure 1 | CRISPR/Cas9-mediated gene editing in *mdx^{4cv}* mice.** (a–e) Strategies for creating a dystrophin mRNA carrying an ORF by removing the *mdx^{4cv}* TAA premature stop codon (the *mdx^{4cv}* C to T point-mutation is depicted in red). (**a**) Strategy 1 (Δ5253) utilizes both dual- and single-vector approaches to target introns 51 and 53 (arrows = sgRNA target sites shown in a 5'-3' direction based on target strand) to direct excision of exons 52 and 53 (**b**). (**c**) Strategy 2 (53*) utilizes a dual-vector approach to target exon 53 on either side of the stop codon, relying on HDR (utilizing a WT DNA template) or NHEJ to generate either full-length WT dystrophin (**d**) or a partial in-frame deletion of exon 53 (**e**).

quantification of NHEJ events resulting from the deletion of both exons 52 and 53 could not be determined via deep sequencing. However, deep sequencing of PCR amplicons generated across the individual target sites could be used to quantify the instances where on-target DNA cleavage did not result in the excision of the intervening 45 kb segment. Using this approach, gene editing efficiencies at introns 51 and 53, respectively, were 8.6% and 8.2% for the dual-vector (Sp) approach and 3.5% and 2.7% for the single vector (Sa) approach (Fig. 2a; Supplementary Fig. 2; Supplementary Table 1). Reverse transcription PCR (RT–PCR) analysis revealed a predominant shorter dystrophin transcript that lacked the sequences encoded on exons 52 and 53 as determined by sequencing of the excised unique band (Fig. 2b,c).

For strategy 2, the combined gene editing efficiency for both target sites within exon 53 was 2.3%, as determined by deep sequencing (Fig. 2d; Supplementary Fig. 3; Supplementary Table 1). Encouragingly, successful HDR was detected in 0.18% of total genomes (Fig. 2d; Supplementary Fig. 4; Supplementary Tables 1 and 2). While this efficiency was low (∼8% of the edited genomes resulted from HDR), the data show that myogenic cells within dystrophic muscles are at least modestly amenable to HDR-mediated dystrophin correction following CRISPR/Cas9 targeting. Analysis of dystrophin transcripts isolated from four treated samples revealed a unique shorter RT–PCR product that, following sequencing of individual cloned RT–PCR products, was shown to correspond to a complete deletion of exon 53 (Supplementary Fig. 3). This unanticipated exclusion of exon 53 from the mRNA likely resulted from larger indel mutations disrupting splicing enhancer signals located within this exon[28]. Successful editing within the main exon 53 RT–PCR product was detected via both T7 endonuclease 1 digestion and Sanger sequencing of individual clones (Supplementary Fig. 3). Deep sequencing of RT–PCR amplicons spanning exons 52 and 53 revealed an

overall editing efficiency of 9.2% at the transcript level with 0.8% of total transcripts corresponding to successful HDR events (Fig. 2d; Supplementary Fig. 3 and 4; Supplementary Tables 1 and 3), thus indicating successful *Dmd* gene editing and HDR within exon 53. Analysis of the sequence reads revealed several types of editing events. For example, 44% (genomic DNA) and 36% (mRNA) of the edited sequences carried insertions, deletions or substitutions that did not shift the reading frame (Fig. 2e). However, only 3% (genomic DNA) and 16% (mRNA) of all edited sequences were in-frame deletions that also removed the *mdx^{4cv}* stop codon. Since ∼8% of all edited genomes and ∼9% of all edited transcripts resulted from HDR (Fig. 2d,e), a total of ∼11% (genomic) and ∼25% (transcript) of the strategy 2 editing events were able to express dystrophin (Fig. 2e, Supplementary Fig. 4; Supplementary Tables 1–3). Overall, on-target editing frequency was significantly higher than for predicted off-target sites sharing the most sequence similarity to the sgRNAs used in strategies 1 and 2 (Supplementary Table 4).

**Induced dystrophin expression improves muscle function.** Establishment of a functional ORF led to significant induction of dystrophin expression in treated TAs as detected by immunostaining of muscle cryosections (Fig. 3a; Supplementary Fig. 5) and by western blotting of whole muscle lysates (Fig. 3b). CRISPR/Cas9-mediated gene correction resulted in full- to near-full-length dystrophin protein expression levels of 0.8–18.6% (dual vector, n = 4) or 1.5–22.9% (single vector, n = 4) for strategy 1 and 1.8–8.4% (53*, dual vector, n = 4) for strategy 2, as compared with wild-type (WT) dystrophin levels (Fig. 3c). In addition to the detection of full- to near-full-length dystrophin, western analysis also revealed a range of shorter dystrophin isoforms (110–160 kD) of unclear therapeutic impact that were more frequent in strategy 2-treated muscles, possibly due to aberrant splicing.

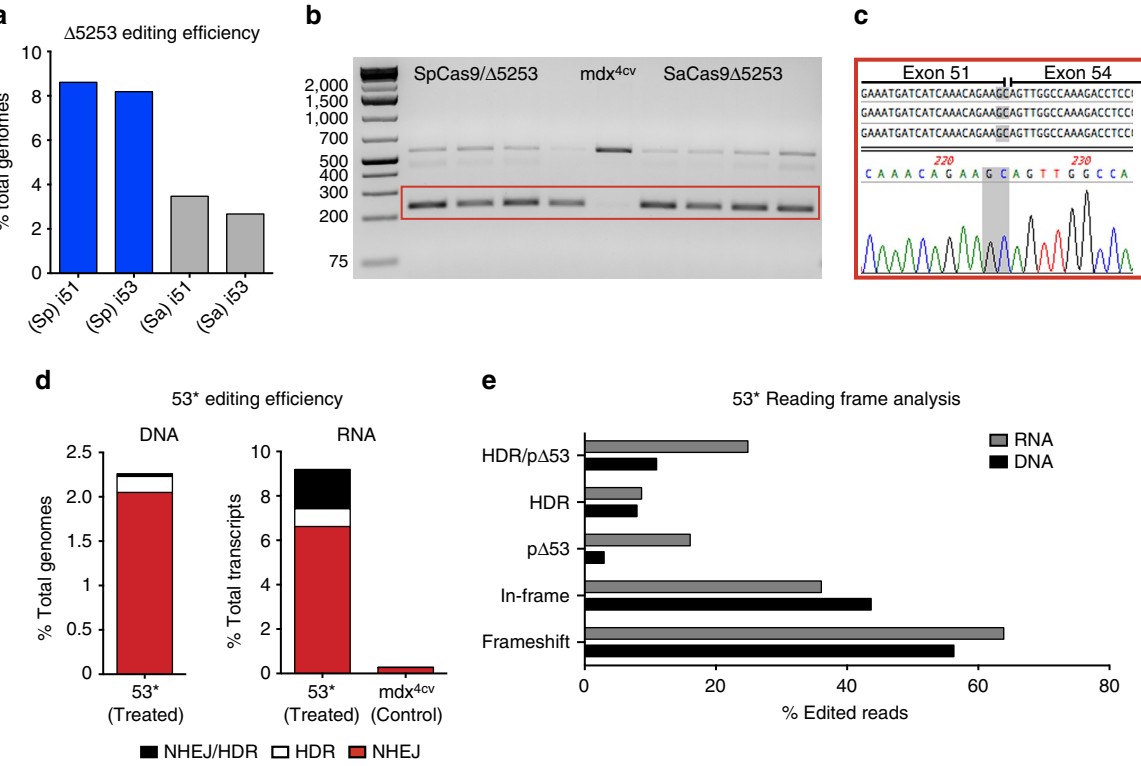

**Figure 2 | _In vivo_ gene editing introduces a functional ORF in _mdx$^{4cv}$_ mouse muscles. (a)** Deep sequencing quantification on PCR amplicons generated from pooled genomic DNA extracted from muscles treated with strategy 1 (Δ5253, $n = 4$), demonstrates successful gene editing at each of the individual target regions. Shown are the percentages of total reads that displayed genomic modifications occurring as a result of NHEJ (including insertions, deletions and substitutions), at sgRNA target sites in introns 51 and 53. **(b)** RT–PCR of target region transcripts isolated from TAs treated with strategy 1 (Δ5253, $n = 4$) showing a predominant shorter product (red box), corresponding to approximately 87.5% of total transcripts based on image densitometry. **(c)** Subclone sequencing of the treatment-specific RT–PCR product (red box in **b**) confirmed that these transcripts lacked the sequences encoded on exons 52 and 53 (the novel junction between exons 51 and 54 is highlighted in grey). **(d)** Deep sequencing quantification of gene editing efficiency on PCR amplicons generated from pooled genomic DNA (left, $n = 5$) and RT–PCR amplicons generated from pooled transcripts (right, $n = 4$) extracted from muscles treated with strategy 2 (53*). Shown are the percentages of total reads that displayed genomic modifications occurring as a result of NHEJ (red), HDR (white) or via a combination of both (black), at both sgRNA target sites in exon 53. **(e)** Deep sequencing reading frame analysis for strategy 2 (53*) shows the percentage of total edited transcript (gray) and genomic (black) reads resulting in frameshift indels, in-frame indels, in-frame deletions without the TAA stop codon (pΔ53), HDR reads (not including mixed NHEJ/HDR reads) and the total percentage of edited reads encoding a functional dystrophin ORF (HDR/pΔ53).

Immunostaining of muscle cross-sections revealed that an average of 41% (Δ5253) and 45% (53*) of myofibers expressed dystrophin (Fig. 3d). Of note, dystrophin-positive myofibers in treated TAs were significantly larger than myofibers of untreated _mdx$^{4cv}$_ controls and than dystrophin-negative fibres within treated muscles (Fig. 3e,g; Supplementary Fig. 6), constituting an average of 54% (Δ5253) and 61% (53*) of the myogenic cross-sectional area with a maximum observed positive area of 68% (Δ5253) and 71% (53*). Dystrophin-positive myofibers within treated muscles also displayed a significant reduction in central nucleation (Fig. 3h).

Induction of dystrophin expression also allowed for sarcolemmal localization of neuronal nitric oxide synthase (nNOS), an important component of the dystrophin-glycoprotein complex that modulates muscle performance (Fig. 4a)[11]. To assess whether CRISPR/Cas9-mediated induction of dystrophin expression would translate into functional improvements we performed _in situ_ measurements of muscle force generation at 18 weeks post-transduction of 2-week-old male _mdx$^{4cv}$_ mice. Encouragingly, the observed dystrophin levels in muscles treated using strategy 1 were maintained at this later time point, resulting in significant increases in specific force generating capacity and protection from contraction-induced injury (Fig. 4b,c).

Conversely, muscles treated according to strategy 2 only displayed a slight but non-significant increase in specific force development, likely due to the lower levels of dystrophin production.

**Systemic delivery induces cardiac dystrophin expression.** On the basis of the higher dystrophin-correction efficiency observed for strategy 1, we proceeded to test this approach following systemic delivery of the AAV nuclease and targeting vectors using a range of doses between $1-10 \times 10^{12}$ v.g. per mouse. Both single- and dual-vector approaches yielded widespread dystrophin expression in the heart, with up to 34% of cardiac myofibers expressing dystrophin at 4 weeks post-transduction (Fig. 5). While both high- and low-vector doses were able to generate dystrophin expression in the heart (Fig. 5b–d), only the high dose was able to generate widespread, albeit variable, dystrophin expression in all muscle tissues analysed (ranging from <10% dystrophin-positive fibres in the quadriceps and EDL muscles to >50% in soleus muscles; Fig. 5e–h). Furthermore, higher cardiac dystrophin expression levels were also obtained with increasing vector dose (Fig. 5i).

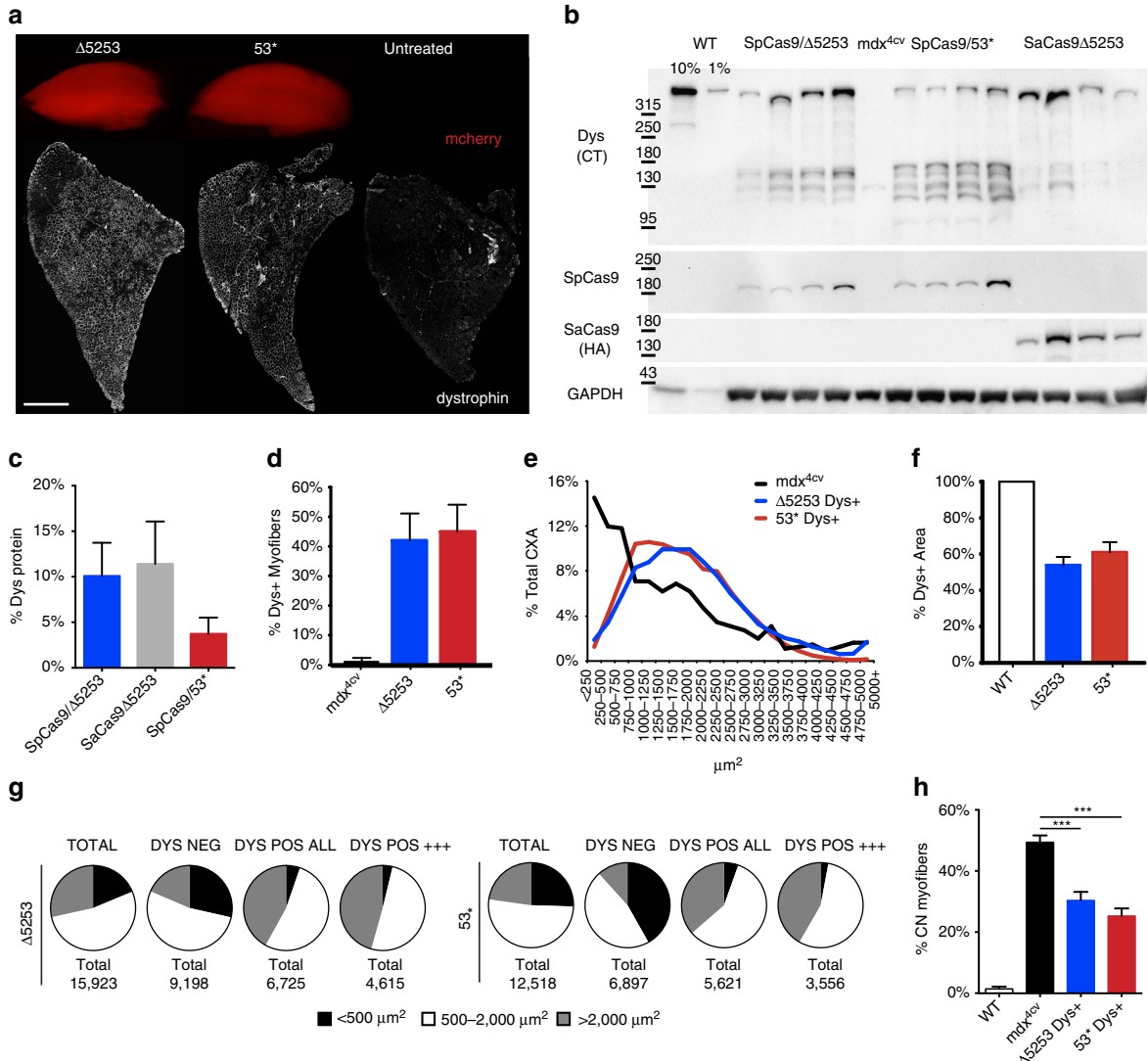

**Figure 3 | Dystrophin expression in treated muscles improves muscle morphology. (a)** TA muscles from treated mice were collected and analysed for expression of the mCherry reporter gene (top) or cryosectioned for immunostaining of dystrophin (bottom). Widespread dystrophin expression resulted from both strategies 1 and 2 (Scale bar, 1mm). **(b)** Western analysis of muscles from treated and untreated mice (WT and $mdx^{4cv}$) showing dystrophin (Dys), SpCas9, SaCas9 and GAPDH expression. Dystrophin was detected using antisera raised against the C terminus (CT); the SaCas9 nuclease carried an HA epitope tag to enable detection with anti-HA antibodies. **(c)** Quantification of GAPDH-normalized dystrophin expression in treated TAs compared with WT muscles ($n = 4$). **(d)** Immunostained cross-sections from treated and control mice were analysed for the percentage of all myofibers expressing dystrophin ($n = 5$). **(e)** Shown is the cross-sectional area (CXA) size distribution of individual myofibers from treated and control muscles ($n > 12,500$ total fibres per group). **(f)** The total myogenic cross-sectional area (CXA) that was dystrophin-positive is shown for treated and WT control muscles ($n = 5$). **(g)** Individual myofiber size distribution for treated TAs relative to dystrophin expression. **(h)** The percentage of myofibers containing centrally located nuclei is shown for dystrophin-positive treated myofibers and for total myofibers of control TA muscles ($n = 5$). Data are shown as mean ± s.e.m. ***$P < 0.001$, (One-way ANOVA multiple comparisons test with Turkey's *post hoc* test).

## Discussion

Our results demonstrate that muscle-specific CRISPR/Cas9-mediated gene editing is effective in inducing dystrophin expression in dystrophic $mdx^{4cv}$ mouse muscles. We also observed localization of dystrophin-associated proteins, such as nNOS, to the sarcolemma and increased muscle force generation. Restriction of Cas9 expression to myogenic cells offers several advantages over ubiquitous expression by preventing expression of the bacterial Cas9 nuclease in non-muscle (including immune effector) cells and eliminating the impact of possible off-target events affecting genes expressed in mitotically active non-muscle cells, such as hepatocytes. Although HDR is believed to occur infrequently in post-mitotic tissues,

at least a fraction of myogenic cells in dystrophic muscles displayed successful HDR-mediated gene correction following CRISPR/Cas9 delivery, as demonstrated by the presence of HDR-derived transcripts. Whether targeting of post-mitotic myonuclei or proliferating myogenic progenitors is responsible for these HDR events is currently unclear. However, MCK regulatory regions are not transcriptionally active in satellite cells or proliferating myoblasts[26,29–31]. In this regard, we previously showed that homologous recombination between separate AAV vector genomes occurs at a moderate frequency in post-mitotic mouse myofibers[32]. Further improvements to HDR-based gene editing strategies could possibly be achieved by inhibiting genes involved in NHEJ[33], and/or via the use of

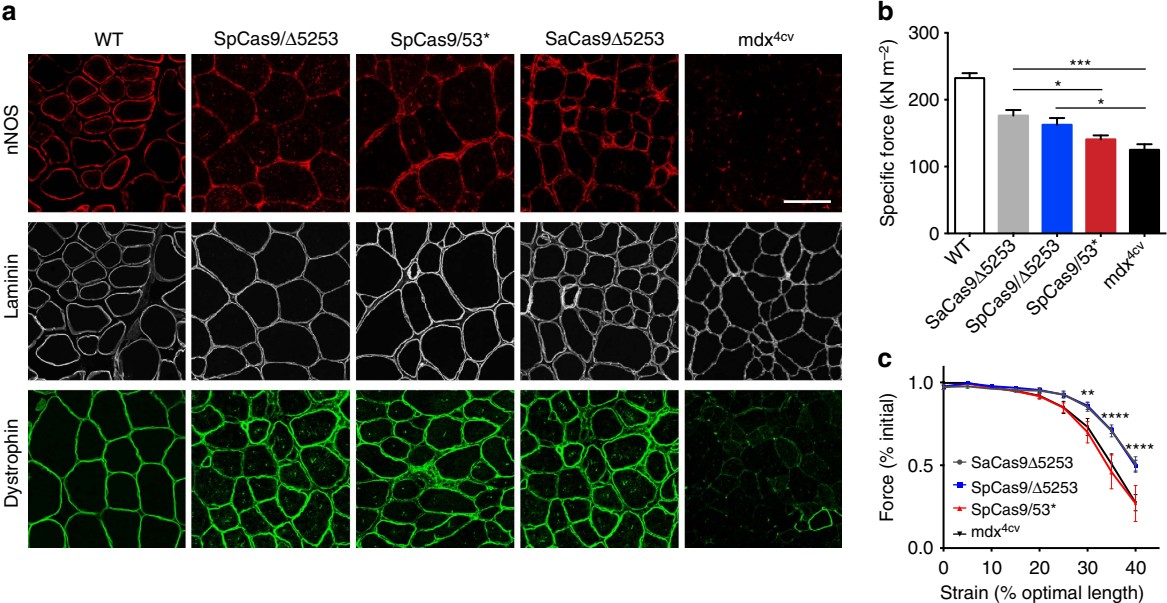

**Figure 4 | CRISPR/Cas9-mediated dystrophin correction localizes nNOS to the sarcolemma and improves muscle function.** (**a**) Immunofluorescent staining for nNOS, laminin and dystrophin in IM-treated and control muscles (Scale bar, 100 μm). (**b**) Specific force generating levels of treated $mdx^{4cv}$ mouse TA muscles 18 weeks post-IM transduction with $2.5 \times 10^{10}$ v.g. of each vector (SaCas9Δ5253 ($n = 8$), SpCas9/Δ5253 ($n = 6$), SpCas9/53* ($n = 8$) and of untreated age-matched WT ($n = 3$) and $mdx^{4cv}$ ($n = 6$) muscles. Bars represent mean ± s.e.m. (*$P < 0.05$, ***$P < 0.001$). (**c**) Protection against eccentric contraction-induced injury as demonstrated by measuring contractile performance immediately before increasing length changes during maximal force production in TA muscles of untreated ($n = 5$) versus IM-treated $mdx^{4cv}$ mice (SaCas9Δ5253 ($n = 8$), SpCas9/Δ5253 ($n = 7$), SpCas9/53* ($n = 8$)). Values are represented as mean ± s.e.m. Statistical significance was determined via multiple Student's $t$-test comparisons, (**$P < 0.01$, ****$P < 0.0001$).

alternative CRISPR associated nucleases (such as Cpf1 or Cas9-nickase)[34,35], which may increase the efficiency of precise gene editing if the HDR events were occurring in mitotically active myogenic precursors.

For excision of exons 52–53, both single- and dual-vector approaches were able to induce dystrophin expression with similar efficiencies, despite an apparent higher frequency of editing with the dual vectors. It is possible that the difference in overall gene editing efficiency stems from a difference in the propensity for indel formation between Sp- and SaCas9 following DNA cleavage at the chosen target sites. For instances when DNA cleavage did not result in deletion of the intervening 45 kb segment, SpCas9 may have generated indels at the cut sites at higher frequencies than SaCas9, resulting in a perceived higher editing efficiency. Actual deletion of the intervening sequence may in fact have been comparable, which the downstream (mRNA and protein) data reflect. Nevertheless, a dual-vector approach currently offers more flexibility in terms of allowing for variations in the ratio between administered nuclease versus targeting components, which may prove important for efficiency. If efficient transduction of myogenic stem cells (satellite cells) can be achieved *in vivo*, dystrophin correction could be permanent by ensuring continued generation of dystrophin expressing myofibers during normal muscle turnover. While our previous results indicated that satellite cell transduction using AAV6, 8 or 9 is very low compared with myofibers[36], one other group found that AAV9 was able to target these stem cells with modest efficiency[15]. The reasons for these differing results are unclear, but significantly greater targeting efficiencies will likely be needed to support long-term regeneration from corrected myogenic stem cells. While the CK8 regulatory cassette in conjunction with CRISPR/Cas9 gene editing is clearly useful for correcting dystrophin mutations in myofibers, CK8 activity

in satellite cells or proliferating myoblasts has not been observed[24,36].

Initial results from CRISPR/Cas9-mediated gene editing strategies are encouraging for the development of future treatments for DMD, but further studies are needed to enhance dystrophin production bodywide, as will be needed to treat or prevent dystrophy in patients[37,38]. Equally important, the effects of potential off-target events will need to be investigated rigorously for each gene editing strategy to ensure short and long-term safety.

## Methods

**Cloning and vector production.** Plasmids containing regulatory cassettes for expression of Cas9 or gRNAs flanked by AAV serotype 2 inverted terminal repeats (ITRs) were generated using standard cloning techniques. The spCas9 nuclease expression cassette was generated by PCR cloning of NLS-spCas9-NLS from LentiCRISPRv1 (ref. 39), and insertion into pAAV (Stratagene) containing the ubiquitous elongation factor-1 alpha short promoter (EFS)[39] (for *in vitro* studies in fibroblasts) or the muscle-specific creatine kinase 8 (CK8) regulatory cassette (RC)[24,26] (for *in vivo* studies). (Sp)sgRNA target sequences were selected using the online software ZiFiT Targeter (http://zifit.partners.org/ZiFiT/) and inserted into pLentiCRISPRv1 following BsmB1 restriction enzyme digestion. Two targeting constructs to work in conjunction with SpCas9 were generated by PCR cloning of the U6-(Sp)sgRNA expression cassette from pLentiCRISPRv1 followed by insertion into pAAV plasmids on either side of a CMV-mCherry expression cassette and a HDR template spanning positions X84575274 to X84576081 of the murine DMD gene cloned from C57BL/6 genomic DNA. The corresponding protospacer adjacent motif (PAM) sites at positions X84575612 (G-A) and X84575639 (G-A) within the HDR template were mutated using PCR-mediated mutagenesis while preserving the encoded amino acids (silent mutations) to eliminate or reduce targeting of the DNA repair template by Cas9. The modified HDR sequence, guide RNA sequences as well as primer sequences for cloning and PCR amplification of genomic DNA and complementary DNA (cDNA) are provided in Supplementary Tables 5–6. The SaCas9 single vector expression cassette was generated by replacing the CMV immediate early enhancer and promoter and the bovine growth hormone poly-adenylation (pA) signal in plasmid #61591 (Addgene)[23] with the CK8 RC and a rabbit beta-globin pA signal, followed by PCR cloning and insertion of a second U6-(Sa)sgRNA expression cassette sequential to the first. (Sa)sgRNA

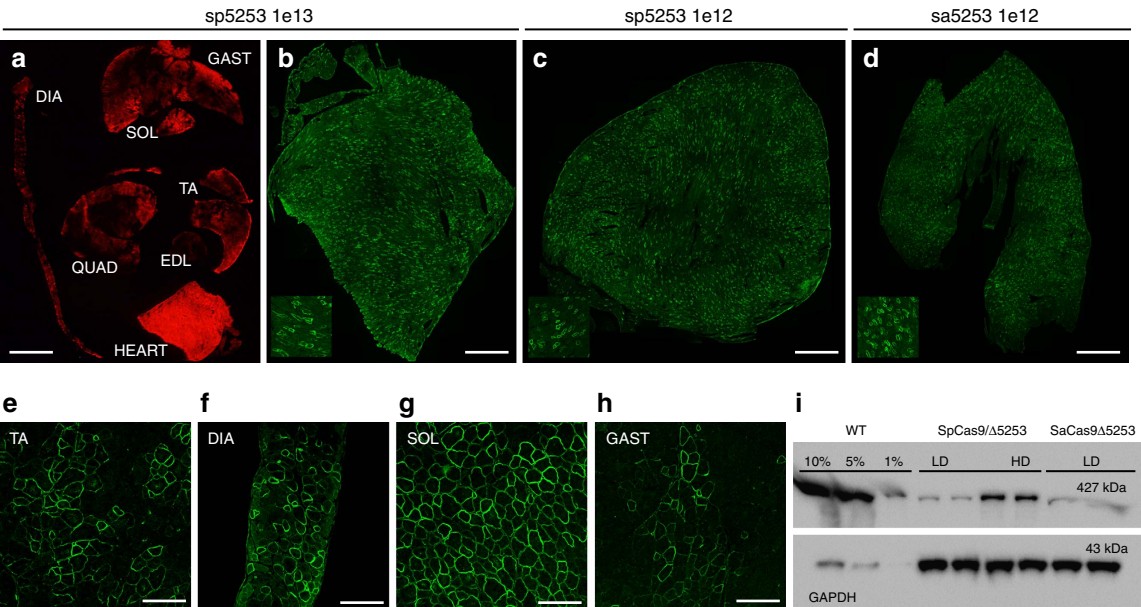

**Figure 5 | Systemic gene editing results in widespread dystrophin expression.** Immunofluorescence analysis of $mdx^{4cv}$ mouse muscles at 4 weeks post systemic transduction with dual (sp5253) and single (sa5253) vector approaches in strategy 1. (**a**) Muscle cross-section showing widespread transduction of multiple muscle groups following high dose ($1 \times 10^{13}/4 \times 10^{12}$ v.g. of nuclease/targeting vectors) dual-vector delivery based on mCherry reporter gene expression, Scale bar, 3 mm. Whole cardiac cross-sections showing dystrophin expression following dual-vector delivery at the high dose (**b**), low dose (**c**, $1 \times 10^{12}/1 \times 10^{12}$) and following single vector delivery at the low dose (**d**, $1 \times 10^{12}$), Scale bars, 1 mm. Insets depict magnified field of views. Widespread but variable dystrophin expression is observed in multiple muscle groups following high dose dual-vector delivery; including TA (**e**), diaphragm (**f**), soleus (**g**) and gastrocnemius (**h**), Scale bars, 100 μm. Western analysis of cardiac lysates demonstrates expression of near full-length dystrophin in low dose (LD) and high dose (HD) treatment groups, with increased dystrophin expression at higher vector doses (**i**).

target sequences were manually selected to target the same locations as the (Sp)sgRNAs and inserted into the U6-(Sa)sgRNA expression cassette via Bsa1 restriction enzyme digestion before inserting the second U6-(Sa)sgRNA cassette into the final construct. Nuclease and targeting pAAV plasmids were co-transfected with the pDG6 packaging plasmid into subcultured HEK293 cells (American Type Culture Collection) using calcium phosphate-mediated transfection to generate AAV6 vectors that were harvested, purified via heparin-affinity chromatography and concentrated using sucrose gradient centrifugation[40]. Resulting titres were determined by Southern analyses using probes specific to the poly-adenylation signal or CMV promoter for nuclease and targeting vectors, respectively.

**Electroporation and culture of primary dermal fibroblasts.** Primary dermal fibroblasts were isolated from 3-week-old male $mdx^{4cv}$ mice[41]. Electroporation of ~600,000 cells per strategy were performed in Invitrogen R-buffer containing 4 μg of both nuclease (EFS-SpCas9)- and targeting (Δ5253/53*) plasmid expression constructs using a Neon transfection system (Invitrogen) with three 10 ms pulses of 1,650 volts. Cells were subsequently seeded on 0.1% gelatin-coated culture vessels and maintained for 12 days in Dulbecco's modified Eagle medium supplemented with Penicillin-Streptomycin, Sodium pyruvate, L-Glutamine and 15% fetal bovine serum (Thermo Fisher Scientific) before harvest and DNA isolation (DNeasy, Qiagen).

**Animals.** All animal experiments were approved by the Institutional Animal Care and Use Committee of the University of Washington. Intramuscular delivery of $2.5–5 \times 10^{10}$ v.g. of each vector (nuclease and targeting) was performed via longitudinal injection into tibialis anterior (TA) muscles of 2–12-week-old male C57BL/6-$mdx^{4cv}$ ($mdx^{4cv}$) mice. For strategy 1, systemic delivery of $1 \times 10^{12}$ v.g. (low dose) to $1 \times 10^{13}$ v.g. (high dose) was performed via retro-orbital injection into 11 week-old male $mdx^{4cv}$ mice ($n = 3$). Both dual- and single-vector approaches were evaluated at the low dose of $1 \times 10^{12}$ v.g. of each vector, while the dual-vector approach was also evaluated at a high dose of $1 \times 10^{13}$ v.g. of the nuclease vector and $4 \times 10^{12}$ v.g. of the targeting vector. The $mdx^{4cv}$ mouse model of DMD harbours a nonsense C to T mutation in exon 53 leading to a loss of dystrophin expression[20]. These mice exhibit ~10-fold lower frequencies of revertant dystrophin expressing muscle fibres than the original $mdx^{scsn}$ mouse strain, which provides much greater assurance that dystrophin-corrected fibres resulted from gene targeting rather than spontaneous reversion.

**Tissue harvest and processing.** Muscles were collected and analysed at 4 weeks post-transduction and compared with age-matched male non-injected $mdx^{4cv}$ and WT mice, except for mice undergoing physiological measurements which were analysed at 18 weeks post-transduction. Medial portions of TA muscles were embedded in Optimal Cutting Temperature (O.C.T.) compound (VWR International) and fresh frozen in liquid nitrogen cooled isopentane for immuno-fluorescence analysis. The remaining portions of TA muscles were snap frozen in liquid nitrogen and ground to a powder under liquid nitrogen in a mortar kept on dry ice for subsequent extraction of DNA, RNA and protein.

**Immunohistochemical and morphometric analyses.** TA cross-sections (10 μm) were co-stained with antibodies raised against alpha 2-laminin (Sigma, rat monoclonal, 1:200) and the C-terminal domain of dystrophin (a kind gift from Dr. Stanley Froehner at the University of Washington Department of Physiology and Biophysics, rabbit polyclonal, 1:500). Serial sections were stained with antibodies raised against neuronal nitric oxide synthase (Invitrogen, rabbit polyclonal, 1:200). Slides were mounted using ProLong Gold with DAPI (Thermo Fisher Scientific) and imaged via Leica SPV confocal microscope at the University of Washington Biology Imaging Facility (http://depts.washington.edu/if/). Confocal micrographs covering the entirety of injected TA muscle sections were acquired and montaged using the Fiji toolset (ImageJ) and Photoshop (Adobe). Quantification of dystrophin-positive myofibers and dystrophin-positive muscle cross-sectional area was performed via semi-automated tracing and measurement of 1,250 to 3,500 individual myofibers per TA using Adobe Photoshop ($n = 5$ TAs per treatment group). Automated quantification of central nucleation was performed using software developed in-house by Rainer Ng (CHAMP) running on the Matlab platform.

**Nucleic acid and protein analyses.** DNA and RNA were isolated using Trizol reagent (Invitrogen) according to the manufacturer's recommendations. Approximately 500 bp amplicons across the targeted regions of genomic DNA were generated by PCR using Phusion proof-reading polymerase (New England Biolabs, NEB) and analysed for targeting efficiency using T7 endonuclease 1 (NEB), next generation sequencing (BGI International or in-house) or Sanger sequencing (Simpleseq, Eurofins MWG Operon) of subclones of PCR amplicons (Zero Blunt TOPO, Invitrogen). The T7 endonuclease assay was performed by denaturing and re-annealing the amplified PCR product followed by treatment with T7 endonuclease 1 to cleave indel-derived heteroduplex PCR products. Analysis of dystrophin-targeted transcripts by RT–PCR of the target regions was performed on

cDNA made using Superscript III first-strand synthesis supermix (Invitrogen). Specific indel mutations or deletions in the dystrophin transcript were identified by Sanger sequencing of individual subclones of RT–PCR fragments. Muscle proteins were extracted in radioimmunoprecipitation analysis buffer (RIPA) supplemented with 5 mM EDTA and 3% protease inhibitor cocktail (Sigma, Cat# P8340), for 1 hour on ice with gentle agitation every 15 min. Total protein concentration was determined using Pierce BCA assay kit (Thermo Fisher). Muscle lysates from WT (10 and 1 µg), untreated $mdx^{4cv}$ (30 µg) and treated $mdx^{4cv}$ (30 µg) mice were denatured at 99 degrees Celsius for 10 min, quenched on ice and separated via gel electrophoresis after loading onto Bolt 4–12% Bis-Tris polyacrylamide gels (Invitrogen). Protein transfer to 0.45 µm PVDF membranes was performed overnight at constant 34 volts at 4 °Celsius in Towbin's buffer containing 20% methanol. Blots were blocked for 1 hour at room temperature in 5% non-fat dry milk before overnight incubation with antibodies raised against the C-terminal domain of dystrophin (Froehner Lab, Rabbit polyclonal, 1:10,000), anti-SpCas9 (Millipore, mouse monoclonal, 1:2,000), anti-HA (Roche, Rat monoclonal-HRP conjugated, 1:2,000) for detection of HA-tagged saCas9 and Gapdh (Sigma, Rabbit polyclonal, 1:100,000). Horseradish-peroxidase conjugated secondary antibody staining (1:50,000) was performed for 1 h at room temperature before signal development using Clarity Western ECL substrate (BioRad) and visualization using a Chemidoc MP imaging system (BioRad). Gel- and blot- band densitometry measurements were performed on unsaturated images using ImageJ software (National Institutes of Health).

**Deep sequencing.** Approximately 200–250 bp PCR products were generated across target-, and the top predicted off-target sites for each sgRNA using Platinum Taq High-Fidelity polymerase (Invitrogen) or Phusion High-Fidelity Polymerase (NEB). Potential off-target sites were identified using ZiFiT Targeter software for SpCas9. CRISPR Rgen tools Cas-OFFinder software (http://www.rgenome.net/cas-offinder/) was used to identify potential off-target sites for saCas9, using a mismatch number of ≤3, DNA bulge size ≤1 and RNA bulge size ≤1. For Strategy B, genomic deep sequencing was performed on a ~230 bp nested PCR product generated from an initial ~500 bp product amplified across exon spanning both target sites. To eliminate false detection of the HDR template DNA present in AAV vectors, the primer pair used to generate the 500 bp product was designed with one primer annealing outside of the region complimentary to the HDR template. The resulting PCR product was isolated following gel electrophoresis (GeneJET gel extraction kit, Thermo Fisher Scientific) before performing nested PCR followed by a second gel extraction. For each site analysed, amplicons from 4–5 mice were pooled and subjected to standard Illumina library preparation (A-tailing, adaptor ligation and amplification using NEBNext library preparation kit (NEB)), and QC'd using a BioAnalyzer before paired end (PE150) sequencing on an Illumina MiSeq system (Illumina Inc., San Diego, CA). Libraries were barcoded for multiplexed sequencing and subsequent reads were parsed and QC'd using custom scripts (Trim galore software (http://www.bioinformatics.babraham.ac.uk/projects/trim_galore/), phred33 score ≥ 30) and standard Illumina tools. On-target paired end (PE150) sequencing of DNA amplicons generated from muscles treated according to strategy 2 (53*) was performed by submitting the samples to BGI International (BGI Americas, Cambridge, MA). Uniquely mapping read pairs were used for downstream analysis using the CRISPResso software pipeline[42]. For CRISPResso analyses: 25 bp at each end of the amplicon were excluded from quantification, the window size around each cleavage site used to quantify NHEJ events was set to 5 bp and sequence homology for an HDR occurrence was set to 98%. Following CRISPResso analysis, manual analysis and quantification was performed by searching for defined sequences in the quality-filtered and adapter-trimmed deep sequencing FASTQ files to provide further information on specific genotypes generated by strategy 2. For DNA reads, search sequences were chosen to span the region containing both target sites and the site of the C–T mutation. For RNA reads, search sequences were defined to span a region starting from within exon 52 (>45 kb away from the target region) extending past the prototypical cut site at the 3′ end of the target region.

**Muscle physiology.** Eighteen weeks post-transduction, treated $mdx^{4cv}$ mice together with age-matched controls were anaesthetized with 2,2,2-tribromoethanol (Sigma) and assayed in situ for force generation[43]. Briefly, a 4–0 silk suture was tied around the distal TA tendon and to a lever attached to a force transducer. After determination of optimal muscle fibre length ($L_0$) the maximum isometric tetanic force was measured during electrical stimulation using Dynamic Muscle Control v5.420 software (Aurora Scientific). Muscle cross-sectional area (CSA) was calculated by dividing muscle mass (mg) by fibre length (mm) and 1.06 mg mm$^{-3}$ (density of mammalian skeletal muscle). Specific force values were obtained by normalizing maximum isometric tetanic force production to CSA. Protection against contraction-induced injury was evaluated by measuring force production during progressive lengthening contractions beyond optimal fibre length[44].

**Statistical analyses.** Data values are represented as mean ± s.e.m. and were analysed in Excel (Microsoft) and Prism6 (GraphPad). Measurements were analysed for statistical significance using one-way analysis of variance (ANOVA) multiple comparison tests with Turkey's post hoc tests unless otherwise stated. Statistical significance was set to $P < 0.05$.

**Data availability.** Sequence data supporting the findings of this study have been deposited in the sequence read archive (SRA) with the BioProject accession code PRJNA358248 (https://www.ncbi.nlm.nih.gov/bioproject/PRJNA358248).

The remaining data are available within the article and its Supplementary Information files and from the corresponding author upon reasonable request. Full scans for western blots are available in Supplementary Fig. 7.

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

## Acknowledgements

We thank Eleanor Chen of the University of Washington Department of Pathology for providing valuable advice regarding CRISPR/Cas9-mediated gene editing, James Allen and Christine Halbert of the Viral Vector Core of the Senator Paul D. Wellstone Muscular Dystrophy Cooperative Research Center for generating AAV vectors, and Ranier Ng and Ladan Mozaffarian for helpful discussions and assistance with CHAMP software. Supported by NIH grants U54AR065139 and R01AR44533, and by grants from the Muscular Dystrophy Association (USA).

## Author contributions

N.E.B. planned and performed experiments, analysed data and drafted the manuscript; J.K.H. gathered experimental data, prepared figures and assisted with manuscript writing; G.L.O. assisted with experiments, provided reagents and helped edit the manuscript; M.P.P. provided reagents and helped design experimental approaches; C.R.A. and R.D.H. assisted with sequence analysis and interpretation of data; S.D.H. provided reagents, assisted with experiments and helped write the manuscript; J.R.C. assisted with experiments and interpreting data and helped write the manuscript; J.S.C. helped design and plan the project, provided reagents, interpreted data and assisted with manuscript preparation and editing. He also assumes overall responsibility for the manuscript and its contents.

## Additional information

**Competing financial interests:** The University of Washington, J.S.C, S.D.H and N.E.B. have a pending patent application on muscle-specific expression of Cas9. The other authors declare no competing financial interests.

DOI: 10.1038/ncomms16007    OPEN

# Corrigendum: Muscle-specific CRISPR/Cas9 dystrophin gene editing ameliorates pathophysiology in a mouse model for Duchenne muscular dystrophy

Niclas E. Bengtsson, John K. Hall, Guy L. Odom, Michael P. Phelps, Colin R. Andrus, R. David Hawkins, Stephen D. Hauschka, Joel R. Chamberlain & Jeffrey S. Chamberlain

Nature Communications 8:14454 doi: 10.1038/ncomms14454 (2017); Published 14 Feb 2017; Updated 23 Jun 2017

This Article contains an error in Fig. 4, for which we apologize. In panel a, the image reporting dystrophin labelling following SaCas9Δ5253 treatment was inadvertently duplicated from the corresponding image following SpCas9/Δ5253 treatment. The correct version of this figure appears below as Fig. 1. The raw data associated with this experiment are provided as a separate Supplementary Data file.

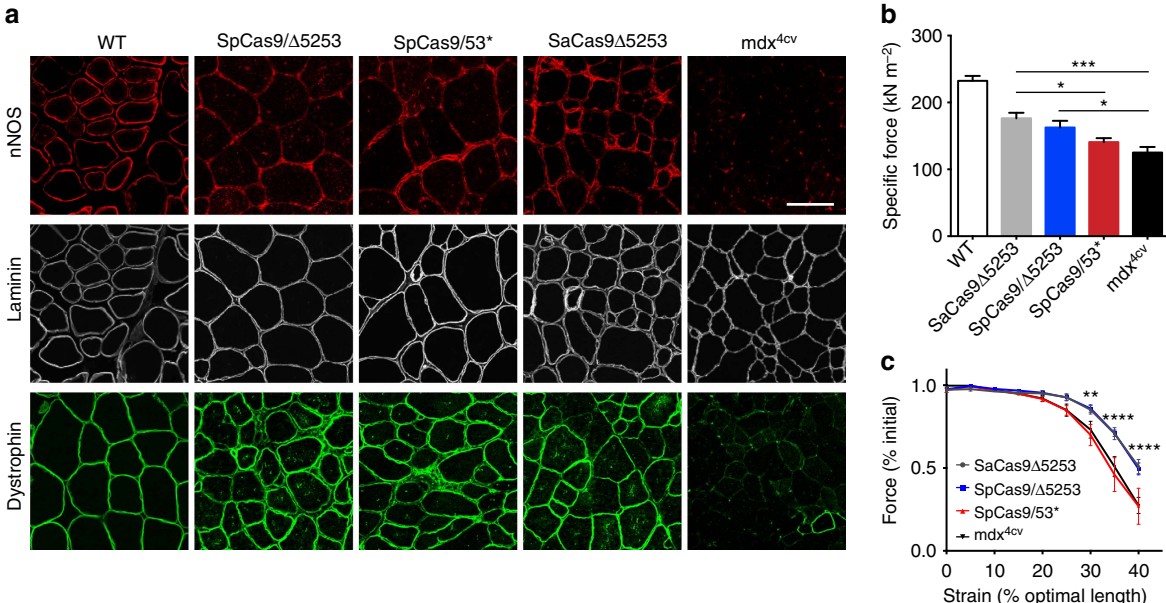

**Figure 1 |**

