## [Peer Review file · Nature Communications]

Reviewers' comments:

Reviewer #1 (Remarks to the Author):

In the manuscript "Enhanced muscle-specific CRISPR/Cas9 editing of the dystrophin gene ameliorates pathophysiology in a mouse model of DMD" Bengtsson and coauthors describe two different CRISPR/Cas9-mediated gene editing approaches to restore an open reading frame of dystrophin protein in mdx4cv mouse model of DMD. In both situations Cas9 expression is driven by a muscle specific CK8 promoter, which is supposed to restrict gene editing events to mature myofibers. Using AAV6-mediated delivery of CRISPR/Cas9 components (intramuscular injection) the authors observed restored expression of dystrophin corresponding to increase in force generation in some experiments.

Unfortunately, the data presented here are only marginally novel (removal of exons to restore an ORF) and in part poorly substantiated by the data (intraexonic deletion + HDR-mediated correction).

Three groups have recently published data regarding restoration of dystrophin ORF in mdx model of DMD by removal of the affected exon 23 using AAV-mediated delivery of CRISPR/Cas9 system. The current study (strategy 1) used a very similar strategy to remove exons 52 + 53, so it does not represent a significant advance in the field.

Specific critical points for strategy 1:

1. The authors mention that the removal of genomic DNA between guides i51 and i53 was quite low. It would be important to quantify the percentage of edited genomes for this deletion to reconcile the data (Figure 1g) with the high proportion of edited transcripts (delta52+53). They need to take into account possible preferential amplification of shorter amplicons that can affect the quantification.
2. Supplementary Figure 1 used for quantification of indel formation for individual guides is of poor quality. Using a different detection system should be used for these calculations. Moreover, for i53 it is not clear to me why the combined sum of the two cleavage products (indicated by arrows) does not correspond to the length of the undigested amplicon.
3. The mutation position distribution of NHEJ for i51 seems odd (suppl. Figure 2), since for most guides it usually peaks around the predicted cleavage site. Is there anything unusual about this particular guide?

Critical points for Strategy 2 (intraexonic deletion + HDR-mediated correction).

1. Homologous recombination is generally considered to be suppressed in postmitotic cells. Since it is the first report claiming to observe occurrence of HDR in mature myofibers, substantially more rigorous approaches are necessary to support this claim. For example, deep sequencing analysis of intraexonic deletion by two guides approach without an HDR template is absolutely essential to take into account possible sequencing errors inherent to NGS platforms. In addition, 5'guide + HDR template and 3' guide + HDR template conditions are also crucial for this analysis.
2. In supplementary Figure 2b there is a plot demonstrating genotypes and frequencies of suggested

HDR events. It is surprising that there is such a large proportion of wt traces, compared to 5'-pHDR and 3'-pHDR, since it would require two recombination events in close proximity to each other. Are the parameters used to run Crispresso algorithm too relaxed? Is there a different way to analyze these frequencies? What would be a distribution of these reads in a negative control sample (untreated mice) or mice treated with two guides without the HDR template?

3. The authors suggest that the CK8 promoter-based Cas9 expression is restricted to mature myofibers to explain HDR based correction. Is it possible that this promoter is "leaky" and Cas9 can be expressed in dividing cells, where an HDR can take place? There are different types of cycling cells present in muscle undergoing degeneration/regeneration, and if HDR happens in these cells, these events can be detected by NGS. Can the authors provide evidence that Cas9 expression is restricted ONLY to myofibers after intramuscular AAV administration.

4. In Figure 1k an estimation of gene editing for the predominant transcript was reported to be 11.2% using T7 endonuclease assay. Is this Figure representative? It is hard to understand why after the digestion, a "smaller" product appears to be more intense than the "larger" product and not the other way around.

5. In addition, it would be important to ascertain whether the HDR-edited transcript is present in the main RT-PCR product. NGS might be suitable to accomplish this.

For both strategies the authors need to address an issue of possible off target effects after CRISPR/Cas9-mediated gene editing and also utilize systemic routes of AAV administration to test an applicability of the approach for the mutation correction in different muscle groups.

Reviewer #2 (Remarks to the Author):

In the manuscript by Bengtsson et al., the authors modify the Dmd gene of the mdx4cv mouse model using the CRISPR/Cas9 system. Although several other groups have previously published CRISPR genome editing of Dmd in mdx mice, this report does offer a few new features, such as the use of a CK8 muscle-specific driver to express Cas9, and the use of mdx4cv instead of mdxScSn mice. Also, they show evidence of Dmd gene correction via homologous recombination in post-mitotic myofibers.

A weakness of this study is that it falls short of offering more pertinent information to further advance this field. There are many variables in the experimental design with insufficient data analysis to give meaningful conclusions or guidance to the field. Most importantly, no experiments were done to assess the expression of Cas9 in the heart by the CK8 muscle-specific driver.

Specific comments:

1. The CK8 promoter was used to enrich muscle delivery but comparison of CK8 and CMV was not done. They rely on previous data generated by other groups to assess efficiency of CK8 versus CMV promoters. In the discussion they write: "While restriction of Cas9 expression to post-mitotic myogenic cells may result in a perceived lower genomic DNA targeting efficiency"...what is meant by "perceived"? It would have been informative if they assessed and measured the efficiency of CK8 and CMV promoters in the same study. In particular, the authors reported CRISPR/Cas9-mediated dystrophin correction resulted in full- to near-full length dystrophin protein expression levels of 0.8-18.6% (dual vector) or 1.5-22.9% (single vector) for strategy 1 and 1.8-8.4% (53*, dual vector) for strategy 2, as compared with WT dystrophin levels. How does this compare to CMV vectors?
2. The "possible" aberrant splicing suggested with strategy 2 is disconcerting. What design change in the strategy (vector or guide) should be made to minimize "unanticipated" effects?
3. DMD patients develop cardiomyopathy. A major weakness of this study is that there is no information about the heart. Is muscle-specific CK8 effective for Cas9 expression in the heart? What is the efficiency of gene editing in the heart using CK8 to express Cas9 in the heart?
4. The results with HDR are not well developed. The extent of HDR in post-mitotic cells remains debatable. The authors have the opportunity to carry out experiments to explore HDR (varying time, age, tissue etc.) but fail to do so in a meaningful way. Based on limited data they report, "successful HDR was detected in 0.18% of total genomes". Is 0.18% meaningful? It is critical to the field to know if HDR is a viable option for muscle. Comprehensive analysis is warranted.
5. In the Methods it is written that the IM delivery was performed into the TA of 2-12 week-old mice. This is a large range in age. It is not clear when different ages were used in the experimental plan. Is there a difference in gene editing efficiency in muscle between 2-week and 12-week old mice? Please discuss and include the age of the mice for each experiment.

Reviewer #3 (Remarks to the Author):

A. The manuscript "Enhanced muscle-specific CRISPR/Cas9 editing of the dystrophin gene ameliorates pathophysiology in a mouse model of DMD" by Bengtson et al describes several strategies to correct dystrophin restoration in the mdx 4cv mouse model for Duchenne muscular dystrophy. AAV-mediated delivery of Cas9 and guide RNAs through intramuscular injection results in genome editing and restoration of dystrophin in the tibialis anterior muscle.

B. The work is a clear furthering of the frontiers of this approach compared to the 3 papers that were published in Science earlier this year in the 'regular' mdx mouse model (Long et al, Nelson et al, Tabebordbar et al, 2016): authors use another mouse model, with a mutation in the mutation hotspot, use muscle specific promoters and use various strategies to bypass the mutation.

E. The authors use a local injection into the TA muscle. While this is sufficient for proof-of-concept, the fact that for DMD systemic treatment is warranted should at least be discussed, especially since this is the major bottleneck for current gene addition therapy approaches for DMD. Genome editing does not solve this issue, it faces the same challenge.

H. The manuscript is at times quite unclearly written. Eg. First paragraph, outlining why flexibility is required: it would be better to swap the frequency and the incidence of spontaneous mutations - the combination is what causes the need for flexibility - the frequency by itself is not necessarily a problem (if all patients have the same mutation it would not be an issue). First outlining the variation in mutations and the high number of new mutations might be more clear. Other examples: line 1 of paragraph 2 of the results is very vague. Same paragraph on the next page, the explanation of how NHEJ could allow dystrophin restoration is not very clear. The model has a nonsense mutation - ergo the reading frame is not disrupted and cannot be restored. NHEJ can remove the mutation while maintaining the reading frame.

Second page of the results when describing the deep sequencing: it would be helpful to add that pcr products were analyzed by deep sequencing - that way it would have been clear to this reviewer why deep sequencing for the del52/53 was not feasible (with deep sequencing on DNA it would have been feasible). Also this part of the result is not very clear - e.g. excluding these events from the analysis? Which events? Those that could not be picked up (these are the events mentioned in the previous sentence)? This makes no sense

E. It is shown that HDR works in the mdx mouse, which is great. However, the authors should discuss that DMD mouse models are characterized by very efficient regeneration. This could be an explanation for the fact that HDR works in dystrophic mouse muscle. There is no guarantee it would also take place in dystrophic human muscle.

C/E/G. Second paragraph of the results section, authors mention that strategy 1 is potentially applicable to a majority of DMD patients with mutations affecting multiple exons or single exons with small mutations. However, in fact exon 52-53 deletion/skipping applies to only 0.3% of patients (Human Mutation 2009, Aartsma-Rus et al). In the discussion they again say this could be applicable to a large number of DMD cases. Targeting this region IS applicable to larger numbers of patients, however, the specific approach used here, applies to a small group of patients.

C/F. RNA analysis is performed with RT-PCR and quantification is done on PCR fragments. This is suboptimal, since most likely the shorter fragments will have an amplification bias (especially with a

large exon like exon 53) leading to an overestimation of deleted events. I am not at all convinced about the percentages mentioned - most likely these are vast overestimations. The authors should either not try to quantify OR use a more quantitative technique.

G/E. Last paragraph of the discussion: the authors state that 20-30% of dystrophin is needed to prevent the debilitating symptoms of DMD. This is outdated information. Recent insight has revealed that patients generating minute amounts of dystrophin leads to a slower disease course (e.g. exon 44 skippable patients, numerous publications on e.g. these patients having a later age of loss of ambulation (work from the Leiden group), slower disease progression as measured with the 6 minute walk test (Mercuri group) or North Star Ambulatory Assessment (Ricotti and Muntoni) and having more dystrophin (work from Muntoni). Having said that, the most important determinant for therapeutic effect is probably time at intervention - restoring 20% dystrophin in a 30 year old patient will likely have less effect than restoring 1% in a one-year old patient. I appreciate that due to space concerns the authors cannot go in detail about this issue - however, I do believe they cannot state the 20-30% of normal is what should be aimed for.

Minor suggestions

7. The official nomenclature states "DMD gene" and "dystrophin protein" (with DMD in italics, DMD for human, Dmd for mouse). Instead the authors use dystrophin gene throughout the manuscript.

8. In the first paragraph after the abstract, the authors discuss that AAV vectors could be lost during myofiber turnover. This phenomenon has actually been reported in the GRMD dog model and the authors should cite this work to back up their statement (work from Luis Garcia, Mol Ther 2012)

9. The deletion fragments in suppl figure 1 are quite faint especially compared to the untargeted fragments - are the authors sure about the percentages of targeting?

10. The authors speculate that the fact that dystrophin positive fibers are larger is due to protection against damage. It is also possible that the larger fibers were taking up the AAV vectors preferentially.

11. Methods: DNA, RNA and protein analyses: manufacturers should be manufacturer's

12. Statistical analysis: p in P-value should be a capital P in italics

13. Figure 1e: is the cross sectional area truly mm² or micrometer²?

14. The authors could consider moving figure 2f and g to the supplementary figures (e.g. add it to suppl fig 5).

Reviewers' comments:

Reviewer #1 (Remarks to the Author):

In the manuscript "Enhanced muscle-specific CRISPR/Cas9 editing of the dystrophin gene ameliorates pathophysiology in a mouse model of DMD" Bengtsson and coauthors describe two different CRISPR/Cas9-mediated gene editing approaches to restore an open reading frame of dystrophin protein in mdx4cv mouse model of DMD. In both situations Cas9 expression is driven by a muscle specific CK8 promoter, which is supposed to restrict gene editing events to mature myofibers. Using AAV6-mediated delivery of CRISPR/Cas9 components (intramuscular injection) the authors observed restored expression of dystrophin corresponding to increase in force generation in some experiments.

Unfortunately, the data presented here are only marginally novel (removal of exons to restore an ORF) and in part poorly substantiated by the data (intraexonic deletion + HDR-mediated correction).

Three groups have recently published data regarding restoration of dystrophin ORF in mdx model of DMD by removal of the affected exon 23 using AAV-mediated delivery of CRISPR/Cas9 system. The current study (strategy 1) used a very similar strategy to remove exons 52 + 53, so it does not represent a significant advance in the field.

Response: The previous groups used a single approach to remove 213 bp of DNA from the mouse genome (exon 23). We used 4 different approaches. In the first 2, we used either a *single vector* system (with SaCas9) or a *dual vector* system (with SpCas9) to remove 45,000 bp of DNA, including exons 52-53. In the third approach, we specifically targeted a mutation in exon 53, rather than removing the entire exon (some dystrophin exons can't be excised and still restore production of a functional protein). Final, we explored HDR as a method to *completely correct* the dystrophin gene. While perhaps none of these 4 approaches are major breakthroughs, we feel they represent a significant advancement from the original approach, especially when applied to a disease that arises from *new mutations anywhere* in the 79 exon, 2.2 MB *DMD* gene.

Specific critical points for strategy 1:

1. The authors mention that the removal of genomic DNA between guides i51 and i53 was quite low. It would be important to quantify the percentage of edited genomes for this deletion to reconcile the data (Figure 1g) with the high proportion of edited transcripts (delta52+53). They need to take into account possible preferential amplification of shorter amplicons that can affect the quantification.

Response: The statement regarding the low efficiency has been removed since it is not feasible to accurately quantitate the very large deletion in strategy 1. However, the presence of large amounts of edited transcripts suggests that the efficiency is significant. PCR analysis across the deleted region has been added to supplemental figure 1, demonstrating the presence of a unique deletion specific product in treated muscles.

Furthermore, repeated deep sequencing analyses on PCR products generated across the individual target sites within introns 51 and 53 on DNA isolated from muscles treated using both single- and dual vector approaches for strategy 1 now shows significant editing of ~8% for the dual vector approach and ~3% for the single vector approach (see new figure 2, supplemental figure 1 and supplemental table 1).

2. Supplementary Figure 1 used for quantification of indel formation for individual guides is of poor quality. Using a different detection system should be used for these calculations. Moreover, for i53 it is not clear to me why the combined sum of the two cleavage products (indicated by arrows) does not correspond to the length of the undigested amplicon.

Response: This figure has been removed since it represented gene editing in transfected fibroblasts *in vitro* in order to demonstrate that our gRNAs worked during initial tests. Quantification done on this figure, which we agree was not great, is more reflective of transfection efficiency than gene editing efficiency. The gene editing efficiency quantifications presented in the new figure 2 is much more reflective of the actual data.

3. The mutation position distribution of NHEJ for i51 seems odd (suppl. Figure 2), since for most guides it usually peaks around the predicted cleavage site. Is there anything unusual about this particular guide?

Response: We have explored this issue by running additional sequencing reactions. Plots generated from repeated deep sequencing demonstrate peak values at the predicted cleavage site (new supplemental figure 1).

Critical points for Strategy 2 (intraexonic deletion + HDR-mediated correction).

1. Homologous recombination is generally considered to be suppressed in postmitotic cells. Since it is the first report claiming to observe occurrence of HDR in mature myofibers, substantially more rigorous approaches are necessary to support this claim.

Response: To support our observations of successful HDR we performed deep sequencing on RT-PCR products generated across exons 52-53, that were derived from RNA isolated from both treated and untreated muscles. This analysis reveals an even greater presence of HDR derived transcripts in treated muscles while none are observed in untreated muscles (see **new figure 2, supplemental tables 1-3 and supplemental figure 3**). Furthermore, the presence of edited dystrophin transcripts supports that muscle specific editing is occurring.

The reviewer also noted that HDR is “considered to be **suppressed** in postmitotic cells”. We agree, and our data show that the HDR approach was the **least efficient** of the methods we tested, but it did lead to clearly detectable HDR, as shown by the sequencing data. Note that we previously published an manuscript documenting homologous recombination in muscle tissue using AAV vectors (Odom GL, Gregorevic P, Allen JA, Finn E and Chamberlain JS: Gene therapy of *mdx* mice with large truncated dystrophins generated by recombination using rAAV6. *Mol Ther* 2011; 19:36-45. PMC3017440) so this is NOT an unprecedented concept. It is perhaps worth noting that AAV vectors are **single-stranded**, which has led to high rates of homologous recombination in many other cell

types (e.g. Chamberlain JR, Schwarze U, Wang PR, Hirata RK, Hankenson KD, Pace JM, Underwood RA, Song KM, Sussman M, Byers PH, Russell DW: Gene targeting in stem cells from individuals with osteogenesis imperfecta. *Science* 2004; 303:1198-1201. (*Targeting Gene Therapy for osteogenesis imperfecta*, Commentary in *New England Journal of Medicine*, 2004; 350:2302-2304).

2. In supplementary Figure 2b there is a plot demonstrating genotypes and frequencies of suggested HDR events. It is surprising that there is such a large proportion of wt traces, compared to 5'-pHDR and 3'-pHDR, since it would require two recombination events in close proximity to each other. Are the parameters used to run Crispresso algorithm too relaxed? Is there a different way to analyze these frequencies? What would be a distribution of these reads in a negative control sample (untreated mice) or mice treated with two guides without the HDR template?

Response: Genotypes of individual sequence reads were not determined using CRISPResso, but rather *via* manual verification of individual sequence reads within the quality filtered and adapter trimmed fastq file, spanning 60 bp across the HDR region from 18 bp upstream of the 5' PAM site to 11 bp downstream of the 3' PAM site). There is a strong possibility that a large proportion of the WT traces represent background since it comprises a single nucleotide substitution. By introducing single nucleotide substitutions (other than T or C) at the *mdx*^{4cv} point mutation site and at other various positions of interest, a level of background can be estimated as shown in supplemental table 2. The same analysis has now been performed on transcripts isolated from both treated and untreated mice to further substantiate the data, as seen in the **new supplemental figure 3 and supplemental table 3**.

3. The authors suggest that the CK8 promoter-based Cas9 expression is restricted to mature myofibers to explain HDR based correction. Is it possible that this promoter is "leaky" and Cas9 can be expressed in dividing cells, where an HDR can take place? There are different types of cycling cells present in muscle undergoing degeneration/regeneration, and if HDR happens in these cells, these events can be detected by NGS. Can the authors provide evidence that Cas9 expression is restricted ONLY to myofibers after intramuscular AAV administration.

Response: CK8 is transcriptionally inactive in all dividing cells tested. While we cannot definitively prove that the HDR occurred exclusively in post-mitotic cells, several lines of evidence strongly support this concept. First, AAV6 displays a very poor ability to infect satellite cells (less than 5% of cells targeted; Arnett LHA, Konieczny P, Ramos JN, Hall J, Odom GL, Yablonka-Reuveni Z, Chamberlain JR and Chamberlain JS: Adeno-associated viral (AAV) vectors do not efficiently target muscle satellite cells. *Mol Ther: Meth Clin Develop.* 2014; 1:14038.) Thus, few of the satellite cells take up vector and could even express Cas9. In contrast, these same vector doses lead to near 100% targeting of post-mitotic myofibers (same paper, and others we've published, such as Gregorevic *et al*, *Nature Medicine*, 2004; 10:828-834). Secondly, the regulatory regions of the MCK gene have been repeatedly shown to be inactive in myoblasts and other cycling cell types (see below in regards to what is known about the MCK enhancer/promoter). Thus, we have

here a combination of a vector that is **poorly tropic for muscle stem cells** combined with a **gene regulatory cassette that is inactive in dividing cells**.

4. In Figure 1k an estimation of gene editing for the predominant transcript was reported to be 11.2% using T7 endonuclease assay. Is this Figure representative? It is hard to understand why after the digestion, a "smaller" product appears to be more intense than the "larger" product and not the other way around.

Response: This figure has been moved to supplemental figure 2 and contains the pooled main RT-PCR product from 4 treated mice (an entire data set). More than one edited product is likely present within this "larger" RT-PCR product since there are 2 target sites located within the product which could give rise to a variety of transcripts resulting from indel formation, various sizes of double-cleavage generated deletions and HDR, leading to a non-uniform intensity distribution. Similar to concerns raised for strategy 1, there may also be preferential retention of corrected transcripts due to nonsense-mediated decay, potentially enhancing certain edited transcripts over others.

5. In addition, it would be important to ascertain whether the HDR-edited transcript is present in the main RT-PCR product. NGS might be suitable to accomplish this.

Response: Deep sequencing of RT-PCR products generated across exons 52-53 has been performed for both treated and untreated mice (see **new figure 2, supplemental tables 1, 3 and supplemental figure 3**).

For both strategies the authors need to address an issue of possible off target effects after CRISPR/Cas9-mediated gene editing and also utilize systemic routes of AAV administration to test an applicability of the approach for the mutation correction in different muscle groups.

Response: Deep sequencing was performed to estimate DNA cleavage at the top predicted off-target sites for each target site. No significant off-target effects were observed (see supplemental table 4).

We also tested strategy 1 at various doses for systemic delivery into male *mdx^{4cv}* mice and observed widespread but variable dystrophin expression throughout all muscle groups analyzed. In particular, substantial dystrophin expression was observed in the heart and highly vascularized muscle groups, such as the soleus (see **new figure 4**).

Reviewer #2 (Remarks to the Author):

In the manuscript by Bengtsson et al., the authors modify the *Dmd* gene of the *mdx4cv* mouse model using the CRISPR/Cas9 system. Although several other groups have previously published CRISPR genome editing of *Dmd* in *mdx* mice, this report does offer a few new features, such as the use of a CK8 muscle-specific driver to express Cas9, and the use of *mdx4cv* instead of *mdxScSn* mice. Also, they show evidence of *Dmd* gene correction via homologous recombination in post-mitotic my fibers.

A weakness of this study is that it falls short of offering more pertinent information to further advance this field. There are many variables in the experimental design with insufficient data analysis to give meaningful conclusions or guidance to the field. Most importantly, no experiments were done to assess the expression of Cas9 in the heart by the CK8 muscle-specific driver.

Specific comments:

1. The CK8 promoter was used to enrich muscle delivery but comparison of CK8 and CMV was not done. They rely on previous data generated by other groups to assess efficiency of CK8 versus CMV promoters. In the discussion they write: "While restriction of Cas9 expression to post-mitotic myogenic cells may result in a perceived lower genomic DNA targeting efficiency"...what is meant by "perceived"? It would have been informative if they assessed and measured the efficiency of CK8 and CMV promoters in the same study. In particular, the authors reported CRISPR/Cas9-mediated dystrophin correction resulted in full- to near-full length dystrophin protein expression levels of 0.8-18.6% (dual vector) or 1.5-22.9% (single vector) for strategy 1 and 1.8-8.4% (53*, dual vector) for strategy 2, as compared with WT dystrophin levels. How does this compare to CMV vectors?

Response: The use of a ubiquitous promoter such as CMV, as opposed to a muscle-specific promoter such as CK8, could lead to gene editing in all cell types transduced by AAV (including fibroblasts, hepatocytes, etc.). This has several unwanted consequences. For one, editing of all cell types in muscle tissue rather than only in myogenic cells would generate a larger fraction of edited genomes per total genomes analyzed, thus creating a **perceived** increase in editing efficiency. For the sake of re-introducing muscle dystrophin expression in DMD, only successful editing of **myogenic** genomes would be functionally relevant. Our approach limits gene editing to myogenic cell nuclei.

More importantly, the CMV promoter is ubiquitously active, such that delivery of Cas9 by AAV would lead to long-term expression of this nuclease in muscle AND non-muscle cells, including mitotically active cells. AAV is especially tropic for liver, thus from a safety stand point why would one want to deliver a nuclease to all cells of the body when the target is post-mitotic myofibers? On a related point, we have been meeting with the FDA about gene therapy trials for DMD, and they pointedly commented that they DO NOT want to see the CMV promoter used again for muscular dystrophy therapies (for one thing CMV is very active in immune effector cells). One AAV human clinical trial was performed that used the CMV promoter (Mendell JR, *et al.*: Dystrophin immunity in Duchenne's muscular dystrophy. N Engl J Med 2010; 363:1429-1437.) and it led to a **cellular immune response against dystrophin** that resulted in rejection of the vector. Thus, for us the question was **not** how MCK compared with CMV, since CMV is not clinically relevant, but whether a **muscle-specific promoter could even work** for gene editing. Our data show editing efficiencies at least as good as in the 3 papers that previously used CMV. In view of these considerations we do not feel there is a justification for comparing MCK to a ubiquitous promoter that has been linked to Cytotoxic T cell-mediated immune responses.

We were confused why the reviewer claimed we relied on "other groups" to compare MCK to CMV. Our co-author, Dr. Hauschka, has published far more in this area than any other lab. The Hauschka lab was the first to clone the MCK gene, they identified

the first muscle-specific enhancer (in the MCK gene), and his lab, with the Chamberlain lab, has published numerous comparisons of CMV vs MCK, including the newest iteration, CK8. CK8 contains modified regulatory components derived from enhancer & promoter portions of the mouse M-creatine kinase gene. Its *in vivo* transcriptional activity in conjunction with AAV systemic delivery has been extensively studied using a wide variety of reporter cDNAs (e.g., Alkaline Phosphatase, GFP, and Luciferase), as well as with numerous therapeutic cDNAs (e.g., micro-dystrophin). CK8 is expressed at high levels in all skeletal muscles as well as in cardiac muscle (see some examples below). In particular, **the figure below** [Hu et al, 2014] provides a demonstration of CK8's activity in **striated** muscle and its lack of activity in non-muscle tissues such as the liver compared to the activity of the ubiquitously expressed CBA (a modified CMV promoter), and Liver Thyroxine-Binding Globulin RCs. Here, one week old mice received 3×10^{12} vector genomes intravenously and were assayed for luciferase expression at 3 weeks. (*The red dot in the head region is jaw muscle.*)

Muscle-Specific Expression of CK8 Regulatory Cassette

Nborn mice: IV inj 3×10^{12} vg/kg: AAV10-CK8-Luciferase Assay @ 3 wks

Here are a select few examples of studies of MCK RCs:

Chamberlain JS, Jaynes JB and Hauschka SD: Regulation of creatine kinase induction in differentiating mouse myoblasts. *Mol Cell Biol* 1985; 5:484-492.

Jaynes, JB, Chamberlain JS, Buskin J and Hauschka SD: Transcriptional regulation of the muscle creatine kinase gene and regulated expression in transfected mouse myoblasts. *Mol Cell Biol* 1986; 6:2855-2864.

Jaynes JB, Johnson JE, Buskin JN, Gartside CL, Hauschka SD: The muscle creatine kinase gene is regulated by multiple upstream elements, including a muscle-specific enhancer. *Mol Cell Biol* 1988; 8:62-70.

Hauser MA, Robinson A, Hartigan-O'Connor D, Williams-Gregory D, Buskin JN, Apone S, Kirk CJ, Hardy S, Hauschka SD, and Chamberlain JS: Analysis of muscle creatine kinase regulatory elements in recombinant adenoviral vectors. *Mol Ther* 2000; 2:16-25.

Hartigan-O'Connor D, Kirk C, Crawford R, Mule J and Chamberlain JS: Immune evasion by muscle specific gene expression in dystrophic muscle. *Mol Ther* 2001; 4:525-533

Salva, M. Z. *et al.* Design of tissue-specific regulatory cassettes for high-level rAAV-mediated expression in skeletal and cardiac muscle. *Mol Ther* 2007; 15:320-329.

Himeda, C. L., Chen, X. & Hauschka, S. D. Design and testing of regulatory cassettes for optimal activity in skeletal and cardiac muscles. *Methods Mol Biol* 2011; 709:3-19.

Hu C, Kasten J, Park H, Bhargava R, Tai DS, Grody WW, Nguyen QG, Hauschka SD, Cederbaum SD, Lipshutz GS. Myocyte-mediated arginase expression controls hyperargininemia but not hyperammonemia in arginase-deficient mice. *Mol Ther* 2014; 10:1792-1802.

2. The "possible" aberrant splicing suggested with strategy 2 is disconcerting. What design change in the strategy (vector or guide) should be made to minimize "unanticipated" effects?

Response: There are not many potential target sites available within small exons, such as exon 53 of the *Dmd* gene, so there are not many alterations that can be made. Furthermore, exon 53 has been shown to contain multiple splice enhancer sequences and it is commonly known that altering splice sites or even deleting whole exons can alter splicing patterns of genes. Thus, for any and each individual tailor-made strategy to restore complete full-length dystrophin expression in DMD one would have to observe the outcomes and determine whether there are detrimental effects. Note that some smaller products were observed with all strategies and both Cas9 enzymes, so this was not an isolated event from a single 'bad' sgRNA. We also note that, for the sake of full disclosure, we went out of our way to show a full size range on our western blots, which was **not done** in any of the 3 previous papers on *mdx* mouse gene editing. Thus, one is not able to determine how many shorter dystrophin products were generated in those previous studies.

3. DMD patients develop cardiomyopathy. A major weakness of this study is that there is no information about the heart. Is muscle-specific CK8 effective for Cas9 expression in the heart? What is the efficiency of gene editing in the heart using CK8 to express Cas9 in the heart?

Response: Both the labs of Eric Olson and Stephen Hauschka have published dozens of manuscripts on the MCK gene and its regulatory elements. The MCK enhancer plus promoter are well known to be **highly active in cardiac muscle**. Nonetheless, we now tested strategy 1 at various doses for systemic delivery into male *mdx*^{4cv} mice and observed widespread but variable dystrophin expression throughout all striated muscle groups analyzed. In particular, substantial dystrophin expression was observed in the heart (up to 35% of cardiomyocytes expressing dystrophin) and highly vascularized muscle groups, such as the soleus (**see new figure 4**).

4. The results with HDR are not well developed.....Is 0.18% meaningful? It is critical to the field to know if HDR is a viable option for muscle. Comprehensive analysis is warranted.

Response: We have added additional sequencing data to address efficiency. Please see the response to reviewer 1, point 3.

5. In the Methods it is written that the IM delivery was performed into the TA of 2-12 week-old mice. This is a large range in age. It is not clear when different ages were used in the experimental plan. Is there a difference in gene editing efficiency in muscle between 2-week and 12-week old mice? Please discuss and include the age of the mice for each experiment.

Response: No apparent difference was observed between injecting 2 or 12 week-old mice. The only treatment group injected at 2 weeks of age involved mice used for measurements of muscle function at 18 weeks post-transduction that received 2.5×10^{10} vg instead of 5×10^{10} vg used for the older mice. Minor differences could possibly include variations in vector genomes injected per muscle weight, and that a 2-week injection would also fall before the crisis period involving muscle necrosis and regeneration that occur in *mdx* mice between 3-8 weeks of age, potentially causing vector dilution.

Reviewer #3 (Remarks to the Author):

A. The manuscript "Enhanced muscle-specific CRISPR/Cas9 editing of the dystrophin gene ameliorates pathophysiology in a mouse model of DMD" by Bengtson et al describes several strategies to correct dystrophin restoration in the *mdx* 4cv mouse model for Duchenne muscular dystrophy. AAV-mediated delivery of Cas9 and guide RNAs through intramuscular injection results in genome editing and restoration of dystrophin in the tibialis anterior muscle.

B. The work is a clear furthering of the frontiers of this approach compared to the 3 papers that were published in Science earlier this year in the 'regular' *mdx* mouse model (Long et al, Nelson et al, Tabebordbar et al, 2016): authors use another mouse model, with a mutation in the mutation hotspot, use muscle specific promoters and use various strategies to bypass the mutation.

E. The authors use a local injection into the TA muscle. While this is sufficient for proof-of-

concept, the fact that for DMD systemic treatment is warranted should at least be discussed, especially since this is the major bottleneck for current gene addition therapy approaches for DMD. Genome editing does not solve this issue, it faces the same challenge.

Response: We appreciate the reviewer's comment that this work extends the previous studies in multiple ways. For systemic gene delivery, we have now added this data. See response to reviewer 2, point 3; see new figure 4

H. The manuscript is at times quite unclearly written. Eg. First paragraph, outlining why flexibility is required: it would be better to swap the frequency and the incidence of spontaneous mutations - the combination is what causes the need for flexibility - the frequency by itself is not necessarily a problem (if all patients have the same mutation it would not be an issue). First outlining the variation in mutations and the high number of new mutations might be more clear. Other examples: line 1 of paragraph 2 of the results is very vague. Same paragraph on the next page, the explanation of how NHEJ could allow dystrophin restoration is not very clear. The model has a nonsense mutation - ergo the reading frame is not disrupted and cannot be restored. NHEJ can remove the mutation while maintaining the reading frame.

Second page of the results when describing the deep sequencing: it would be helpful to add that pcr products were analyzed by deep sequencing - that way it would have been clear to this reviewer why deep sequencing for the del52/53 was not feasible (with deep sequencing on DNA it would have been feasible). Also this part of the result is not very clear - e.g. excluding these events from the analysis? Which events? Those that could not be picked up (these are the events mentioned in the previous sentence)? This makes no sense

Response: Thank you for these comments, they are very helpful. We have tried to address the concerns about parts of the manuscript being unclear and hopefully the mentioned parts will now make more sense.

E. It is shown that HDR works in the mdx mouse, which is great. However, the authors should discuss that DMD mouse models are characterized by very efficient regeneration. This could be an explanation for the fact that HDR works in dystrophic mouse muscle. There is no guarantee it would also take place in dystrophic human muscle.

Response: There is no guarantee that anything done in a mouse will work in DMD patients. As noted above, the HDR efficiency was low, but it was detectable and we felt it was worth at least showing the data.

C/E/G. Second paragraph of the results section, authors mention that strategy 1 is potentially applicable to a majority of DMD patients with mutations affecting multiple exons or single exons with small mutations. However, in fact exon 52-53 deletion/skipping applies to only 0.3% of patients (Human Mutation 2009, Aartsma-Rus et al). In the discussion they again say this could be applicable to a large number of DMD cases. Targeting this region IS applicable to larger numbers of patients, however, the specific approach used here,

applies to a small group of patients.

Response: We agree entirely, the relevant sections were not worded as clearly as they should have been. We were trying to say that the ability to target multiple exons would enable application to a large number of patients, not that targeting exon 53 *per se* would be widely applicable. This point became a bit confusing because we were also trying to point out that exon 53 is within the central deletion-prone region of the human *DMD* gene, we did not mean to imply that exon 53 was the deletion prone region all by itself. Our main point was that unlike the previous studies that only focused on exon 23, it will be important to be able to target multiple regions of the gene, so we looked at exons located within the major deletion hot spot. Since there is not a mouse model with a mutation of every exon, we used the 4cv mouse as a model. In any case, we have now re-worded a number of points and hopefully they are now clear and accurate.

C/F. RNA analysis is performed with RT-PCR and quantification is done on PCR fragments. This is suboptimal, since most likely the shorter fragments will have an amplification bias (especially with a large exon like exon 53) leading to an overestimation of deleted events. I am not at all convinced about the percentages mentioned - most likely these are vast overestimations. The authors should either not try to quantify OR use a more quantitative technique.

Response: We recognize that gel densitometry measurements of RT-PCR products are not completely accurate and that there may be a bias for more efficient amplification of shorter products. However, there also appears to be preferential retention of corrected transcripts due to avoidance of nonsense mediated transcript decay as shown and mentioned in Nelson et. al's publication in Science. The quantification provided is meant to serve as an estimate for the relative efficiency of the system to generate corrected transcripts. Further, there is an interesting relationship between the amount of edited transcripts and the amount of dystrophin produced when comparing strategies 1 and 2, where an average of corrected mRNA levels of ~85% translates into expression of up to 22.9% of WT dystrophin levels while 25% of edited transcripts for strategy 2 (based on RT-PCR deep sequencing) translates into dystrophin expression of up to 8.4% of WT.

G/E. Last paragraph of the discussion: the authors state that 20-30% of dystrophin is needed to prevent the debilitating symptoms of DMD. This is outdated information. Recent insight has revealed that patients generating minute amounts of dystrophin leads to a slower disease course (e.g. exon 44 skippable patients, numerous publications on e.g. these patients having a later age of loss of ambulation (work from the Leiden group), slower disease progression as measured with the 6 minute walk test (Mercuri group) or North Star Ambulatory Assessment (Ricotti and Muntoni) and having more dystrophin (work from Muntoni). Having said that, the most important determinant for therapeutic effect is probably time at intervention - restoring 20% dystrophin in a 30 year old patient will likely have less effect that restoring 1% in a one-year old patient. I appreciate that due to space concerns the authors cannot go in detail about this issue - however, I do believe they cannot state the 20-30% of normal is what should be aimed for.

Response: We agree. The 20-30% number came from our work and that of others in mdx mice, and was based on levels needed to fully **prevent** dystrophy from developing. However, the reviewer is correct that there is a huge difference between preventing or curing DMD and alleviating some of the phenotype. Hopefully the new phrasing is better.

Minor suggestions

7. The official nomenclature states "DMD gene" and "dystrophin protein" (with DMD in italics, DMD for human, Dmd for mouse). Instead the authors use dystrophin gene throughout the manuscript.

Response: The phrase 'dystrophin gene' is commonly used in the literature and in our opinion is more clear than using a gene name that is the same as the disease name. Nonetheless, we now adhere to this odd convention in the revised paper.

8. In the first paragraph after the abstract, the authors discuss that AAV vectors could be lost during myofiber turnover. This phenomenon has actually been reported in the GRMD dog model and the authors should cite this work to back up their statement (work from Luis Garcia, Mol Ther 2012)

Response: It is unclear to us that the Garcia paper actually reflects this point, since subsequent papers from that same group (and several others) DID NOT SEE a loss of expression over time. We have discussed this issue multiple times with the Paris group and they do not have an explanation for their contradictory results, thus we have not cited that paper in this context. It is perhaps worth noting that the loss of vector genomes seen by Garcia et al was in a single animal.

9. The deletion fragments in suppl figure 1 are quite faint especially compared to the untargeted fragments - are the authors sure about the percentages of targeting?

Response: This figure has been removed since it represented gene editing in fibroblasts *in vitro* simply to demonstrate that our gRNAs worked during initial tests. The gene editing efficiency quantifications presented in the new figure 2 are much better.

10. The authors speculate that the fact that dystrophin positive fibers are larger is due to protection against damage. It is also possible that the larger fibers were taking up the AAV vectors preferentially.

Response: Perhaps, this is all speculative. We have not seen evidence that larger or smaller fibers display a preference for vector uptake, and that would be a difficult, albeit interesting, point to examine.

11. Methods: DNA, RNA and protein analyses: manufacturers should be manufacturer's

12. Statistical analysis: p in P-value should be a capital P in italics

13. Figure 1e: is the cross sectional area truly mm² or micrometer²?

Response: These errors have been corrected.

Reviewers' comments:

Reviewer #1 (Remarks to the Author):

In the revised manuscript the authors present new experimental data to address several concerns brought up by the reviewer. 1) Systemic administration of CRISPR/Cas9 components (strategy 1) led to widespread but variable expression of dystrophin in skeletal and cardiac muscle of treated mice, suggesting relevance of this approach beyond local intramuscular administration of the virus. 2) Deep sequencing of the Dmd exon 53 RT-PCR amplicons in mice treated with HDR repair cassette (strategy 4) provided additional evidence that HDR events are detectable in skeletal muscle after CRISPR/Cas9-induced homologous recombination. While some of the new data improved the manuscript, interpretation of others as well as removal of some panels from the original Figures raise a few additional concerns.

1. The authors have decided to remove the original Figure used for quantification of indel formation for individual guides citing low transfection efficiencies of fibroblasts as a limiting step. However, analysis of guides' activity is an important prerequisite step for rational choice of guides for gene editing applications, especially in situations where two guides are used together for generation of targeted deletions. To circumvent the problem with low transfection efficiency in fibroblasts, activity of guides can be assessed in an easy to transfect mouse cell line (e.g. Neuro2A).

2. The relevance of the new Figure 2a is unclear, since it does not address editing efficiencies of generating deletions between two guides of interest, but simply reflects proportion of residual modified genomes at each individual site. In their response the authors suggest that "it is not feasible to accurately quantitate the very large deletion in strategy 1". However, it can be accomplished by using ddPCR technique that allows detection and quantification of multiple targets simultaneously. An accurate quantification of genomes with intended deletion is especially important for strategies 1 and 2, since those ones have resulted in increased muscle force generation.

3. For the analysis of off target sites (Supplementary Table 4), every single site that the authors have interrogated so far has a detectable level of sequence variation. Though, individually, these sites do not account for significant "editing efficiency" (last column), taken together they would account for substantial degree of off target activity. It should be considered to interrogate the off target sites in untreated mice as well to infer whether the "editing efficiency" is a result of a bona fide off target activity or sequencing errors.

Minor points:

A. In supplementary Figure 1b, what is the nature of a very prominent high molecular weight band only present in mice treated with strategy 1. Does this strategy generate an additional and preferential gene editing effect?

B. How off target sites were chosen is unclear. This information needs to be added to the Materials and Methods.

C. Most of the outputs for deep sequencing result indicate that manual annotation have been used to determine the frequencies of certain reads. Given the nature of the next generation sequencing prone to errors, detailed description of the exact steps, that were taken for manual selection of reads presented in Supplementary Tables 2 and 3, is necessary. For example, if an HDR sequence is

detected, and there was an additional nucleotide mismatch in surrounding sequence, was this read included in the analysis? Is there a way to analyze the data in an unbiased way using a custom script with defined parameters?

D. The last part of the following statement from the abstract is misleading. "Treated muscles demonstrated production of near- to full-length dystrophin in up to 70% of the myogenic cross-sectional area and a significant increase in force generation following intramuscular delivery." An increase in muscle force generation has not been demonstrated for the HDR-based strategy; the statement needs to be corrected accordingly.

Reviewer #2 (Remarks to the Author):

The revised manuscript is improved by the added experiments and textual revisions. However, the conclusions regarding HDR results require additional experiments to support the conclusions.

Major points (in order of occurrence in the manuscript):

1. The authors show differences in the editing efficiency between single and dual vectors. However, the efficiency of correction of muscle dystrophin expression and specific force measurements are similar. The authors should discuss this point and provide an explanation.
2. Figure 4d: In this study the authors compared the dual vector strategy (using spCas9) and single vector strategy (using saCas9). The authors should follow through and be complete by presenting results for systemic delivery of the saCas9-single vector strategy, in terms of efficiency of correction in cardiac muscle using immunohistochemistry and western blot analysis. The differences are not only single versus double vector strategy but also using different Cas9 species with different PAM sequences. Therefore, it would be informative to provide all the comparison information.
3. Regarding strategy 2 (53*): Since HDR-mediated editing generally does not occur in post-mitotic cells, which exit the cell cycle shortly after birth, such as myofibers, the authors should consider all the controls and possibilities to convince the reader of their conclusions. The authors are missing an important control for strategy 2. The authors should use the gRNA-5' and gRNA-3' vector without the HDR-template to exclude the possibility that the appearance (with 0.18% frequency) of the CAA instead of TAA is not occurring through over DSB repairing or spontaneous mutations. This would seem to be particularly important since this gene has already been described as a target of frequent spontaneous mutation and this region is one of the hot spot regions of the dystrophin gene.
4. "Further improvements to HDR-based gene editing strategies could possibly be achieved by inhibiting genes involved in NHEJ, and/or via the use of alternative CRISPR associated nucleases (such as Cpf1 or Cas9-nickase) (33,34), which may increase the efficiency of precise gene editing." The authors should rephrase this statement since HDR-mediated editing generally does not occur in post-mitotic cells and the reference that they cite for the inhibition of genes involved in NHEJ was performed in proliferating mammalian cells lines.

Reviewer 1.

1. *The authors have decided to remove the original Figure used for quantification of indel formation for individual guides citing low transfection efficiencies of fibroblasts as a limiting step. However, analysis of guides' activity is an important prerequisite step for rational choice of guides for gene editing applications, especially in situations where two guides are used together for generation of targeted deletions. To circumvent the problem with low transfection efficiency in fibroblasts, activity of guides can be assessed in an easy to transfect mouse cell line (e.g. Neuro2A).*

While this is good advice for the future, the reviewer refers to an 'important pre-requisite step for rationale choice of guides'. We showed data previously on our testing, but the figure was not beautiful. After consultation with the editors, we are adding it back as a supplementary figure. Please also note that we had to employ 6 different guide RNAs in this study, and the choice of target sequences was fairly minimal considering the location of some of the targets and the availability of PAM sequences and recognition sequences with minimal off-target predictions. Similarly, the previous Cas9 papers in Science focused on a single choice of sgRNAs to target a specific splice site region of exon 23. The main issue here is that we did the selection, and the data on dystrophin expression clearly show that the guides we chose work well, based on robust dystrophin expression. Thus, while validating guides is a nice precursor to improving target efficiency *in vivo*, our results show that we were able to achieve high *in vivo* targeting efficiency with the guides we selected. We also note that the activity of a sgRNA when assayed by plasmid transfection into a neural cell line *in vitro* will not necessarily predict its activity *in vivo* in muscle. Furthermore, even by doing such a comparison it does not necessarily provide robust comparative data on future studies, as all such experiments would need to be performed in parallel with controls to ensure that identical transfection efficiencies were obtained in such a cell line. For all these reasons we feel that our *in vivo* IM and IV delivery data validate the choice of sgRNAs.

2. *The relevance of the new Figure 2a is unclear, since it does not address editing efficiencies of generating deletions between two guides of interest, but simply reflects proportion of residual modified genomes at each individual site. In their response the authors suggest that "it is not feasible to accurately quantitate the very large deletion in strategy 1". However, it can be accomplished by using ddPCR technique that allows detection and quantification of multiple targets simultaneously. An accurate quantification of genomes with intended deletion is especially important for strategies 1 and 2, since those ones have resulted in increased muscle force generation.*

While we agree that performing this quantification would be interesting, our lab does not currently have access to a digital droplet PCR machine. We are looking into acquiring a ddPCR machine for future studies, but here we did the best we could by looking at the 2 cut sites on either side of the 45 kb region that was deleted. We did quantify deletion efficiency at the mRNA level and at the protein level, which is the ultimate goal of the approach. Critically, we see robust dystrophin expression.

3. *For the analysis of off target sites (Supplementary Table 4), every single site that the authors have interrogated so far has a detectable level of sequence variation. Though, individually, these sites do not account for significant "editing efficiency" (last column), taken together they would account for substantial degree of off target activity. It should be considered to interrogate the off target sites in untreated mice as well to infer whether the "editing efficiency" is a result of a bona fide off target activity or sequencing errors.*

The degree of off-target cutting *per se* is comparable to all other published studies. In this manuscript we were not trying to solve the problem of off-target cutting, and many newer enzymes are arising that may have lower off-target efficiencies. For now our analysis was detailed and robust, and we show the data that was obtained by extensive sequencing of the highest predicted off target sequences. This is the data we were asked for in the previous reviews, and we provided it. An elaboration regarding whether off-target efficiency represents bona fide off target efficiency or sequencing errors was provided in the legend for **supplementary Table 4**. While the vast majority of sequence variations at ON target sites consisted of insertions and deletions (quantification of insertions, deletions and substitutions is also generated by the CRISPResso software pipeline), the much lower frequency of overall sequence variations (editing efficiency) at off target sites consisted mainly of single nucleotide substitutions (indicative of sequencing errors). Furthermore, cDNA sequencing across exon 53 of untreated samples yielded a comparable level of background (~0.3%), as shown in supplementary table 1.

Minor points:

A. In supplementary Figure 1b, what is the nature of a very prominent high molecular weight band only present in mice treated with strategy 1. Does this strategy generate an additional and preferential gene editing effect?

This was a PCR background issue due to attempts to amplify a 45 kb target.

B. How off target sites were chosen is unclear. This information needs to be added to the Materials and Methods.

We used standard methods available online and have added more details to the methods section.

C. Most of the outputs for deep sequencing result indicate that manual annotation have been used to determine the frequencies of certain reads. Given the nature of the next generation sequencing prone to errors, detailed description of the exact steps, that were taken for manual selection of reads presented in Supplementary Tables 2 and 3, is necessary. For example, if an HDR sequence is detected, and there was an additional nucleotide mismatch in surrounding sequence, was this read included in the analysis? Is there a way to analyze the data in an unbiased way using a custom script with defined parameters?

We provided a great deal of additional data and descriptions of how we analyzed the sequence data, and prominent members of the UW Genome Sciences Sequencing Center assisted us. We have provided additional details on the selection of search sequences in the materials and methods section.

D. The last part of the following statement from the abstract is misleading. "Treated muscles demonstrated production of near- to full-length dystrophin in up to 70% of the myogenic cross-sectional area and a significant increase in force generation following intramuscular delivery." An increase in muscle force generation has not been demonstrated for the HDR-based strategy; the statement needs to be corrected accordingly.

Well, this was true for the other strategies tested, and strikes us as an important point. The HDR approach demonstrated a trend towards increased force, but it did not rise to the level of significance. We have re-worded the sentence to remove the word significant.

Reviewer 2:

1. The authors show differences in the editing efficiency between single and dual vectors. However, the efficiency of correction of muscle dystrophin expression and specific force measurements are similar. The authors should discuss this point and provide an explanation.

It is quite possible that the difference in overall gene editing efficiency stems from a difference in the propensity for indel formation between Sp- and SaCas9 following DNA cleavage at the chosen target sites. For instances when DNA cleavage did not result in deletion of the intervening 45kb segment, Sp cas9 may have generated indels at the cut sites at higher frequencies than SaCas9. Thus, resulting in a perceived higher editing efficiency. Actual deletion of the intervening sequence may in fact have been very comparable, which the downstream data seem to reflect. Furthermore, previous transgenic mouse and AAV delivery data from our lab and others shows a tendency for dystrophin levels to accumulate due to mRNA and protein stabilization effects.

2. Figure 4d: In this study the authors compared the dual vector strategy (using spCas9) and single vector strategy (using saCas9). The authors should follow through and be complete by presenting results for systemic delivery of the saCas9-single vector strategy, in terms of efficiency of correction in cardiac muscle using immunohistochemistry and western blot analysis. The differences are not only single versus double vector strategy but also using different Cas9 species with different PAM sequences. Therefore, it would be informative to provide all the comparison information.

We have moved the systemic delivery data to a new figure and added results from IF and western blot analyses from the use of a reduced dose and the SaCas9 single vector. This experiment was not requested in the original reviews, but we provided new data and have now provided even more new data in Figure 4.

3. Regarding strategy 2 (53): Since HDR-mediated editing generally does not occur in post-mitotic cells, which exit the cell cycle shortly after birth, such as myofibers, the authors should consider all the controls and possibilities to convince the reader of their conclusions. The authors are missing an important control for strategy 2. The authors should use the gRNA-5' and gRNA-3' vector without the HDR-template to exclude the possibility that the appearance (with 0.18% frequency) of the CAA instead of TAA is not occurring through over DSB repairing or spontaneous mutations. This would seem to be particularly important since this gene has already been described as a target of frequent spontaneous mutation and this region is one of the hot spot regions of the dystrophin gene.*

The spontaneous appearance of CAA versus TAA is controlled for by accounting for the two introduced silent PAM site mutations that are located at 4 and 23 bp upstream and downstream of the "C", respectively. As can be seen by deep sequencing at the RNA level (Supplementary Table 3), the predominant HDR genotype is A-C-A (where the A's represent the induced PAM site mutations. In order to be truly detected as HDR, a combination of these three specific substitutions must occur. This is further evidenced by the complete lack of genotypes containing 2 or more of these substitutions in untreated mdx control samples (Supplementary Table 3). An indicator of background or spontaneous reversion from T to C could be extrapolated from the detection of C "only" in untreated mdx control samples.

Furthermore, we have elaborated in the discussion that the detected HDR may be occurring in perfusion myocytes, which are abundant in regenerating dystrophic muscles and may be more capable of HDR. A case could be made that detection of HDR in post-mitotic myofibers could be the result of a "cut-and-paste" event, where both the genomic DNA and corresponding sequence in the vector genome are cleaved and inserted into the genome.

The efficiency of this occurring would likely be very low due to the mutated PAM sites present in the vector genome. Additionally, since SpCas9 cleaves target DNA at the PAM -3 position in the vast majority of cases, the likelihood of observing the A-C-A genotype would be close to 0 since the 3'-A would sit well outside of this segment. The most likely genotype generated from a "cut-and-paste" event would be A-C-G (where G is the unaltered nucleotide), and as seen in supplementary Tables 2 and 3, this genotype is detected at much lower frequencies.

The reviewer seems surprised that we detected HDR with a promoter active only in post-mitotic cells. Because of their concerns, we added considerable data in the revision and the text showing that MCK is active in striated muscles, but not in mitotically active cells. We also clarified the sequencing data to make clear that multiple sequence changes were observed that could not arise from simple single base changes. We did the HDR studies in such a way as to distinguish between a simple reversion of the mutation, vs. a double stranded break followed by insertion of the homology template, vs. HDR repair. The data clearly show that HDR was detected. We did add considerable discussion in the text and in the legends to the supplementary figures to clarify these issues, much of this is also in the many pages of supplementary material that we added in response to the first reviews. The important point is that we emphasized in the manuscript that the HDR seen is a relatively rare event, less efficient than the other strategies employed. However, the low level of HDR observed is interesting and we thought we should show the data in case others want to pursue such approaches in attempts to improve the efficiency.

4. "Further improvements to HDR-based gene editing strategies could possibly be achieved by inhibiting genes involved in NHEJ, and/or via the use of alternative CRISPR associated nucleases (such as Cpf1 or Cas9-nickase) (33,34), which may increase the efficiency of precise gene editing." The authors should rephrase this statement since HDR-mediated editing generally does not occur in post-mitotic cells and the reference that they cite for the inhibition of genes involved in NHEJ was performed in proliferating mammalian cells lines.

The reviewer seems to be arguing that HDR can ONLY occur in mitotically active cells, but then argues we should not cite data from mitotically active cells. We did refer in the manuscript to our previous studies showing that homologous recombination can be detected in myofibers following dual AAV delivery (reference 32). We have clarified this point in the discussion.

REVIEWERS' COMMENTS:

Reviewer #2 (Remarks to the Author):

The manuscript is now acceptable for publication.